# ❄SNOWFL: Efficient and Heterogeneous Federated Learning with SNip-OWen-values

## Abstract

Cross-device federated learning often faces heterogeneous clients. These clients carry data with very different values for training high-performance, generalized global models, calling for effective contribution estimation mechanisms. Width scaling with thinner subnetworks and depth scaling via early exits enable participation for heterogeneous clients but still suffer from (i) noisy aggregation across mismatched subnetworks, (ii) under-trained deep layers when few clients reach them, and (iii) costly, client-isolated contribution estimates. We propose SNOWFL, which pairs server-side single-shot pruning at initialization pruning (SNIP) with coalition-structured Owen valuation. SNIP uses a small public, unlabeled set to score connections by loss sensitivity and produce layer-consistent width masks per tier aligned with fixed early exits. During training, we estimate client contributions by first computing Owen values for coalitions and then allocating credit within each coalition via update alignment and diversity. These contribution estimates will be used in both weighted aggregation and drive capacity-aware reassignment. We prove nonconvex convergence to stationarity and, under strong convexity on the retained subspace, linear convergence to a neighborhood. Under matched FLOPs and parameter budgets, SNOWFL achieves state-of-the-art accuracy on vision and language benchmarks, improving strong heterogeneous baselines by up to 15%, while valuation remains data-free except for the small public samples used once for initialization.

## 1 Introduction

Federated learning (FL) trains a single global model across many clients without sharing raw data McMahan et al. (2017). In cross-device settings, client hardware ranges from GPUs to memory/compute-constrained phones and sensors Li et al. (2020b); Karimireddy et al. (2020); Li et al. (2021b). Standard methods such as FedAvg assume one common model on every client; in practice, the weakest devices cannot host or train it and are dropped. Systematic exclusion slows convergence and introduces selection bias, leaving data from weaker clients under-represented Li et al. (2020b); Karimireddy et al. (2020). Training a single downsized model that everyone can handle is not a remedy either, since a one-size-fits-all network typically underfits richer data on capable clients and sacrifices accuracy Diao et al. (2020). Our goal is to engage all clients without degrading the global model by assigning each client a compatible subnetwork and periodically regrouping clients by estimated contribution, so participation and capacity are driven by utility rather than hardware alone.

A common response to heterogeneity is dynamic model scaling. Width scaling assigns thinner subnetworks to constrained clients Diao et al. (2020). Depth scaling equips backbones with early exits so shallower models run on weaker devices Liu et al. (2022a); Kim et al. (2022). Hybrids such as ScaleFL combine both and often add cross-exit distillation to align representations Ilhan et al. (2023). Knowledge transfer via distillation or contrastive objectives further helps large and small models share information Zhu et al. (2021); Li et al. (2021a). These techniques enable heterogeneous participation.

Important gaps remain. Averaging width-pruned models can misalign parameters, and naive width subsetting can underperform simply excluding weak clients Diao et al. (2020). Multi-exit networks alleviate channel mismatch, but separate heads may compete without careful coordination; deep layers can be under-trained because only strong clients traverse them Kim et al. (2022); Ilhan et al. (2023); Lee et al. (2024). Subnetworks and exits also accumulate different BatchNorm statistics, which destabilizes aggregation; turning off BN tracking avoids drift but often reduces accuracy. Thus, while scaling enables participation, it can compromise optimization stability or rely on heavy distillation.

A complementary lever is pruning at initialization. Single-shot network pruning (SNIP) scores each connection at initialization by the loss gradient with respect to a binary mask and keeps the most salient channels in one pass Lee et al. (2018). Examples of SNIP-based methods include GraSP and SynFlow Wang et al. (2020a); Tanaka et al. (2020). In FL, a server-side SNIP step can define data-aware, layer-consistent masks for each submodel so clients train aligned, task-relevant subnetworks rather than ad hoc slices. A brief BatchNorm calibration on public or unlabeled data then harmonizes statistics along pruned and exit paths, stabilizing training and evaluation.

Fair and efficient training also requires weighting clients by the utility of their updates. The Shapley value provides an axiomatic notion of marginal contribution but is intractable at FL scale Ghorbani & Zou (2019). Approximations such as ShapFed and GTG-Shapley reduce cost but largely treat clients independently Tastan et al. (2024); Liu et al. (2022b). In practice, clients naturally cluster into a few *coalitions* that share submodel configurations. The Owen value generalizes Shapley to such coalition structures by first valuing groups as a quotient game and then allocating value within each group. We adopt this perspective to compute group- then member-level contributions and to regroup clients by measured utility over rounds, distinct from specific sampling estimators such as FedOwen KhademSohi et al. (2025).

We propose SNOWFL, an Owen–value-based contribution weighting for heterogeneous tiers, complemented by a single-shot, server-side *SNIP* step that produces task-aware, exit-compatible width masks and a brief BN calibration for stability. Together, these pieces stabilize aggregation across heterogeneous subnetworks and improve the accuracy, efficiency and fairness trade-off. We list our contributions below:

- We formalize contribution estimation using the Owen value Owen (1977) over the coalition structure induced by tier assignments: (i) group-level Shapley on the quotient game to value coalitions; (ii) within-coalition allocation that rewards global alignment and non-redundancy among client updates. These scores drive *both* aggregation weights and capacity-aware reassignment, improving stability and convergence under heterogeneity Ghorbani & Zou (2019); Tastan et al. (2024); Liu et al. (2022b); KhademSohi et al. (2025).

- We derive task-aware, layer-consistent width masks aligned with fixed early exits *once* at initialization without the need for client data. Then a brief BN calibration harmonizes statistics along pruned paths. This reduces subnetwork mismatch and keeps aggregation simple Lee et al. (2018); Diao et al. (2020).

- We evaluate SNOWFL against various heterogeneous FL baselines, including HeteroFL, DepthFL, ScaleFL, InclusiveFL, and ReeFL, on vision and language benchmarks under non-iid partitions. SNOWFL improves accuracy by up to 15% relative. Ablations isolate the effects of SNIP pruning and Owen value weighting.

## 2 Related Work

### 2.1 Federated Learning with Heterogeneous Models

Width scaling via subnetwork training (e.g., HeteroFL) enables resource-aware participation but can suffer from parameter mismatch and biased pruning when subnetworks are formed naively Diao et al. (2020). Depth scaling through early exits (InclusiveFL, DepthFL) assigns shallower models to weak devices and aggregates layer-wise, often coupled with distillation

to align shallow and deep representations Liu et al. (2022a); Kim et al. (2022). ScaleFL unifies width+depth scaling, adding multi-exit classifiers and cross-exit self-distillation for consistent aggregation Ilhan et al. (2023). ReeFL refines multi-exit training by sharing a unified classifier across exits to mitigate conflicting objectives and uses dynamic self-distillation Lee et al. (2024). Complementary FL optimizers (FedProx, SCAFFOLD, FedNova) reduce client drift and normalize aggregation in heterogeneous networks but do not natively resolve model-size heterogeneity Li et al. (2020b); Karimireddy et al. (2020); Wang et al. (2020b). Batch normalization personalization (e.g., FedBN) mitigates feature shift by keeping BN locally Li et al. (2021b), but it does not address the subnetwork and exit mismatch created by width and depth scaling; our lightweight calibration is complementary and architecture aware. Beyond width/depth sharing, *knowledge-transfer* routes enable heterogeneous architectures to collaborate without strict parameter alignment: FedMD distills via a public set across disparate models Li & Wang (2019), and FedGKT transfers knowledge between large and small models through group distillation He et al. (2020). For nested subnetworks, FjORD uses ordered dropout to yield consistent, width-scaled submodels that aggregate cleanly Horvath et al. (2021). Finally, large-scale benchmarking such as FedScale highlights practical, system-level heterogeneity patterns and evaluation protocols complementary to algorithmic proposals Lai et al. (2022). In contrast, SNOWFL generates *task-aware subnetworks* via SNIP at initialization and avoids iterative pruning or heavy distillation while preserving a common parameterization for aggregation Lee et al. (2018).

## 2.2 Early-Exit Architectures

Classic early-exit networks such as BranchyNet Teerapittayanon et al. (2016), MSDNet Zhang et al. (2022), and Shallow-Deep Networks Lei et al. (2020) reduce inference cost by exiting confidently at intermediate layers. FL variants (DepthFL, ScaleFL, ReeFL) adapt these ideas to cross-device training with multi-exit heads and inter-exit knowledge transfer Kim et al. (2022); Ilhan et al. (2023); Lee et al. (2024). Our method adopts a *simple, fixed* set of exits for compatibility but focuses the novel contributions on (i) *SNIP-guided width selection* per resource group and (ii) *Owen-based contribution weighting and regrouping*; this separates efficiency (architecture) from fairness (valuation) without adding complex exit policies.

## 2.3 Pruning at Initialization

PaI methods rank parameters/units by saliency at initialization and prune in one shot. **SNIP** computes connection sensitivity to the loss Lee et al. (2018); GraSP preserves gradient flow via a Hessian-based criterion Wang et al. (2020a); SynFlow avoids data dependence by maximizing synaptic flow Tanaka et al. (2020). Unlike iterative pruning, PaI minimizes retraining overhead. In FL, PaI can predefine compatible sparse subnetworks across clients. SNOWFL uses SNIP *server-side* with a small public set, producing layer-consistent masks for groups (width) and designated exits (depth) before training. Complementarily, LotteryFL shows that lottery-ticket subnetworks can be found and exploited in federated training for personalized, communication-efficient models, further motivating one-shot sparsification in FL Li et al. (2020a).

## 2.4 Client Contribution Evaluation and Fairness in FL

Shapley-value-based approaches (e.g., Data Shapley Ghorbani & Zou (2019)) provide principled attributions but are expensive in FL. ShapFed computes class-wise Shapley to drive weighted aggregation Tastan et al. (2024); GTG-Shapley accelerates estimation via guided truncation Liu et al. (2022b); ShapleyFL treats FL as a sequential cooperative game to adapt weights robustly Sun et al. (2023); SPACE estimates contributions in a single round using knowledge amalgamation and prototypes Chen et al. (2023). Secure protocols enable private Shapley computation in cross-silo settings Zheng et al. (2022). In VFL, VerFedSV and surveys discuss fair and efficient vertical contribution measurement Fan et al. (2022); Cui et al. (2024). FedOwen introduces *Owen sampling* to reduce variance and budget in client valuation and to guide adaptive selection KhademSohi et al. (2025). Orthogonal to Shapley-based scoring, Agnostic FL (Q-FFL) formalizes worst-case fairness objectives under client

shifts Mohri et al. (2019), while clustered-diverse client sampling improves exploration–exploitation in selection under heterogeneity Fraboni et al. (2021). These insights motivate SNOWFL's alignment-aware utility, Owen-style group allocation, and contribution-weighted aggregation/regrouping.

## 3 PRELIMINARIES

### 3.1 FEDERATED LEARNING AND HETEROGENEOUS DEPTH/WIDTH

Let $N$ clients minimize the standard FL objective

$$F(\boldsymbol{w}) = \sum_{i=1}^{N} \frac{n_i}{\sum_{j=1}^{N} n_j} \mathbb{E}_{(\mathbf{x}, \mathbf{y}) \sim D_i} \big[ \mathcal{L}(\boldsymbol{w}; \mathbf{x}, \mathbf{y}) \big], \tag{1}$$

with $\boldsymbol{w} \in \mathbb{R}^d$. FedAvg alternates local SGD on selected clients and weighted averaging McMahan et al. (2017); Li et al. (2020b). Under system heterogeneity, clients may train *scaled* submodels while sharing a common parameterization. We distinguish two orthogonal scaling axes: **width scaling** removes channels/neurons to respect compute or memory budgets; **depth scaling** equips the backbone with early-exit heads and allows truncated forward/backward passes so weaker devices stop earlier Diao et al. (2020); Liu et al. (2022a); Kim et al. (2022); Ilhan et al. (2023); Lee et al. (2024). Aggregation aligns the shared parameters of the underlying backbone; exit heads map intermediate features to the common prediction task.

When the backbone admits multiple exits, we train with a standard multi-exit objective

$$\mathcal{L}_{\text{multi}}(\boldsymbol{w}; \mathbf{x}, \mathbf{y}) = \sum_{b=1}^{B} \lambda_b \, \ell\big(h_b(f_{\leq d_b}(\mathbf{x}; \boldsymbol{w})), \mathbf{y}\big), \qquad \lambda_b \geq 0, \ \sum_{b=1}^{B} \lambda_b > 0, \tag{2}$$

where $f_{\leq d_b}$ denotes the backbone truncated at depth $d_b$ and $h_b$ is the associated head. The choices of $B$, exit placements $\{d_b\}$, and coefficients $\{\lambda_b\}$ are architecture-level hyperparameters fixed outside the theory; they are specified with the models in Section 5. Throughout, client sampling follows the protocol in Section 4, and the aggregation rule is contribution-weighted and privacy preserving.

### 3.2 CLIENT VALUATION

Given a utility $\nu(S)$ for any coalition $S \subseteq \{1, \dots, N\}$, the Shapley value

$$\phi_i = \sum_{S \subseteq \{1, \dots, N\} \setminus \{i\}} \frac{|S|! \, (N - |S| - 1)!}{N!} \big[ \nu(S \cup \{i\}) - \nu(S) \big] \tag{3}$$

allocates contributions in a way that uniquely satisfies efficiency, symmetry, dummy, and additivity Ghorbani & Zou (2019). Exact computation is exponential, motivating estimators and structure-exploiting variants in FL Tastan et al. (2024); Liu et al. (2022b); Sun et al. (2023). The Owen value extends Shapley to a priori coalition structures, enabling stratified valuation that first attributes mass to groups and then divides within groups according to within-group signals.

SNOWFL adopts this two-level perspective: round-wise, we evaluate group contributions in a quotient game and perform an intra-group allocation that respects efficiency while remaining *data-free*. Concretely, the utility employed later in Section 4.3 depends only on model updates and their alignment with the aggregated direction; no client examples or labels are accessed. A small public or unlabeled pool is used solely for initialization-time saliency scoring and BN calibration, keeping valuation strictly privacy preserving.

# 4 METHODOLOGY

## 4.1 OVERVIEW

SNOWFL couples a *server-side, one-shot* pruning-at-initialization stage with a *round-wise, coalition-structured* client valuation mechanism. Phase I constructs, for each resource tier, a task-aware width mask that is compatible with a designated early-exit depth, obtained via SNIP on a small public or unlabeled set (no private data). Phase II performs federated optimization over these pruned, multi-exit architectures while computing Owen-style client contributions each round; these contributions govern aggregation weights and the reassignment of clients to tiers. A lightweight batch-normalization (BN) calibration aligns statistics to the current model structure using only public or unlabeled data. See Algorithm 1 for the full SNOWFL training loop.

## 4.2 PHASE I: SALIENCY-GUIDED WIDTH PRUNING AT INITIALIZATION

**Saliency and privacy-preserving scoring.** Let the model have parameters $\boldsymbol{w} \in \mathbb{R}^m$ and binary masks $\boldsymbol{c} \in \{0,1\}^m$ defining a pruned subnetwork $\boldsymbol{w}' = \boldsymbol{c} \odot \boldsymbol{w}$. On a small, fixed public/unlabeled set $\mathcal{D}_{\mathrm{valid}}$ used *only* at initialization (and later BN calibration), we define connection saliencies

$$s_j = \left| \frac{\partial \mathcal{L}(\boldsymbol{c} \odot \boldsymbol{w}; \mathcal{D}_{\mathrm{valid}})}{\partial c_j} \right|\Bigg|_{\boldsymbol{c}=\boldsymbol{1}} = \left| \langle \nabla_{\boldsymbol{w}} \mathcal{L}(\boldsymbol{w}; \mathcal{D}_{\mathrm{valid}}), \boldsymbol{e}_j \odot \boldsymbol{w} \rangle \right|. \tag{4}$$

Thus $s_j$ measures the instantaneous loss sensitivity to removing parameter $w_j$ at initialization. No client-local examples or labels are ever touched by Phase I.

**From parameter scores to tier-consistent channel masks.** Saliencies are aggregated to unit-level quantities $s_k^{(l)}$ (filters/channels or neurons) within each layer $l$; this respects structured pruning and preserves tensor shapes. Given a per-layer budget $\kappa_g^{(l)}$ for tier $g$, we keep the $\kappa_g^{(l)}$ most salient units:

$$\mathcal{I}_g^{(l)} = \mathrm{TopK}\big(\{s_k^{(l)}\}_{k=1}^{K^{(l)}}; \kappa_g^{(l)}\big), \qquad c_{k,g}^{(l)} = \mathbb{I}\{k \in \mathcal{I}_g^{(l)}\}, \tag{5}$$

and enforce cross-layer channel consistency (e.g., pruning an output channel in layer $l$ implies pruning the corresponding input channel in $l+1$). Skip connections are handled by pruning aligned branches so residual additions remain shape-compatible; BN parameters follow their associated channels.

**Geometric compute schedule across tiers.** We order tiers so that $g=1$ is the *full* model, and impose a multiplicative compute budget $\rho \in (0,1)$, with

$$F_g = \rho^{g-1} F_1,$$

where $F_g$ is the target FLOPs for tier $g$ and $F_1$ is the FLOPs of the full backbone with final exit. In Phase I we jointly choose per-layer budgets $\{\kappa_g^{(l)}\}$ and an exit depth $d_g$ so that the estimated FLOPs of the masked, truncated network $f_{\leq d_g}(\cdot; \boldsymbol{w} \odot \boldsymbol{c}_g)$ satisfies $\mathrm{FLOPs}(\boldsymbol{c}_g, d_g) \leq F_g$. When multiple $(\{\kappa_g^{(l)}\}, d_g)$ meet the constraint, we pick the *deepest feasible* exit $d_g$ and let width pruning absorb the compute reduction; this matches practice where stronger pruning permits later exits at the same budget. The pair $(\boldsymbol{c}_g, d_g)$ is computed once and frozen.

**Depth compatibility with early exits.** Exit placements $\{d_b\}$ are fixed at the architecture level. For tier $g$, pruning is applied only up to $d_g$ so that the designated exit receives a well-formed representation; layers deeper than $d_g$ are inactive for that tier but remain available to higher tiers. Phase I yields a ladder $\{(\boldsymbol{c}_g, d_g)\}_{g=1}^{G}$ of task-aware subnetworks, all subgraphs of one global model. Masks and the geometric FLOPs targets $\{F_g\}$ are computed once and then frozen for the remainder of training.

**Why SNIP here.** SNIP's first-order criterion provides a stable, *data-light* proxy for parameter importance at initialization, aligning with our privacy constraints and avoiding multiple costly retraining cycles. Because masks are derived from a single model (not per-client fine-tuned copies), they align channels across tiers, simplifying aggregation in Phase II.

### 4.3 Phase II: Owen-Style Valuation, Weighted Aggregation, and Tier Reassignment

**Masked updates and a geometry of progress.** At round $t$, selected client $i$ trains only within its assigned masked subspace, returning $\Delta \boldsymbol{w}_{i,t}$ with zeros on pruned coordinates. Let

$$\boldsymbol{v}_t = \frac{\sum_{i \in \mathcal{S}_t} \alpha_{i,t} \, \Delta \boldsymbol{w}_{i,t}}{\left\| \sum_{i \in \mathcal{S}_t} \alpha_{i,t} \, \Delta \boldsymbol{w}_{i,t} \right\|_2} \tag{6}$$

be the normalized aggregate direction with provisional nonnegative weights $\alpha_{i,t}$ (e.g., uniform or proportional to $n_i$). We measure contribution using a purely *data-free* utility

$$U_t(A) = \sum_{i \in A} \Big( \max\{\langle \Delta \boldsymbol{w}_{i,t}, \, \boldsymbol{v}_t \rangle, 0\} \Big)^{1/2}. \tag{7}$$

Geometrically, $\langle \Delta \boldsymbol{w}_{i,t}, \boldsymbol{v}_t \rangle = \|\Delta \boldsymbol{w}_{i,t}\|_2 \cos \theta_{i,t}$ is the signed projection of $i$'s update onto the global target. Clipping at zero ignores antagonistic directions; the square root is a concave tempering that reduces winner-take-all effects while still rewarding alignment. Because all $\Delta \boldsymbol{w}_{i,t}$ are represented in the same ambient space (with zeros on out-of-tier coordinates), inner products are well-defined across heterogeneous tiers.

**Tier-level (quotient-game) Shapley with efficiency.** Let $\mathcal{P}_t = \{P_{1,t}, \ldots, P_{G,t}\}$ be the partition of participants by current tier. Defining $U_t^{\mathrm{grp}}(\mathcal{Q}) = U_t\big(\bigcup_{g \in \mathcal{Q}} P_{g,t}\big)$ for $\mathcal{Q} \subseteq \{1, \ldots, G\}$, we estimate each tier's Shapley value

$$\phi_{g,t} = \mathbb{E}_\pi \big[ U_t^{\mathrm{grp}}\big(\mathrm{Pred}_\pi(g) \cup \{g\}\big) - U_t^{\mathrm{grp}}\big(\mathrm{Pred}_\pi(g)\big) \big], \tag{8}$$

by Monte Carlo permutations of tiers, clamping negative increments to zero. We then rescale so $\sum_{g=1}^G \phi_{g,t} = U_t^{\mathrm{grp}}(\{1, \ldots, G\})$, ensuring *efficiency*. If $P_{g,t} = \varnothing$ at round $t$, we set $\phi_{g,t} = 0$.

**Within-tier allocation: alignment and diversity.** Owen's within-coalition division is guided by two signals. The first is global alignment $a_{i,t} = \max\{\langle \Delta \boldsymbol{w}_{i,t}, \boldsymbol{v}_t \rangle, 0\}$. The second is a peer-diversity term computed via within-tier cosine similarities:

$$d_{i,t} = 1 - \frac{1}{|P_{g,t}| - 1} \sum_{j \in P_{g,t} \setminus \{i\}} \frac{\langle \Delta \boldsymbol{w}_{i,t}, \, \Delta \boldsymbol{w}_{j,t} \rangle}{\|\Delta \boldsymbol{w}_{i,t}\|_2 \, \|\Delta \boldsymbol{w}_{j,t}\|_2}, \tag{9}$$

with standard stabilization (small $\varepsilon$ in denominators; if $|P_{g,t}| = 1$ then $d_{i,t} = 1$). Intuitively, clients that explore complementary directions (high $d_{i,t}$) add robustness; clients tightly clustered around the same direction share credit. We combine signals using within-tier min–max normalization $\mathrm{norm}(\cdot)$,

$$z_{i,t} = (1 - \gamma_t) \, \mathrm{norm}(d_{i,t}) + \gamma_t \, \mathrm{norm}(a_{i,t}) + \alpha_t \, \mathrm{norm}(\log n_i), \tag{10}$$

forming soft weights $w_{i,t} = \exp(z_{i,t}) / \sum_{j \in P_{g,t}} \exp(z_{j,t})$ and allocating $v_{i,t} = \phi_{g,t} w_{i,t}$. This respects efficiency within each tier ($\sum_{i \in P_{g,t}} v_{i,t} = \phi_{g,t}$) while balancing aligned progress and exploratory diversity.

**Aggregation and reassignment with capacity constraints.** The global model is updated by

$$\boldsymbol{w}_{t+1} = \boldsymbol{w}_t + \eta_t \sum_{i \in \mathcal{S}_t} \beta_{i,t} \, \Delta \boldsymbol{w}_{i,t}, \qquad \beta_{i,t} \propto v_{i,t} \, \tilde{\alpha}_{i,t}, \quad \sum_i \beta_{i,t} = 1. \tag{11}$$

At scheduled intervals, clients are reassigned by sorting the most recent available $v$ (participants use $v_{i,t}$; non-participants carry forward their last estimate) and partitioning into $G$ tiers under fixed capacity constraints. This prevents collapse to the deepest configuration and yields a stable resource allocation. Reassignment uses the frozen masks $(\boldsymbol{c}_g, d_g)$ from Phase I, so clients move between *compatible* subgraphs without architectural churn.

**Warm-up, smoothing, and stability.** A brief warm-up can avoid noisy early valuations: during the first few rounds one may (i) aggregate uniformly, (ii) accumulate stable estimates of $\boldsymbol{v}_t$, and (iii) delay the first reassignment. Thereafter, applying a short exponential moving average to $\{v_{i,t}\}$ before normalization reduces oscillations without biasing across tiers. These stability controls are architectural-agnostic and reported with training schedules in Section 5.

### 4.4 Batch Normalization Calibration

Pruned pathways and early exits induce heterogeneous activation statistics. To mitigate BN mismatch without private data, we refresh BN buffers either *server-side* (forward passes on $\mathcal{D}_{\text{valid}}$ after aggregation) or *client-side* (brief forward-only passes on local unlabeled data before training when public data are unavailable or shifted). The calibration budget (samples/iterations; which exits are refreshed) is fixed per experiment and listed in Section 5.

### 4.5 Computation and Privacy

**One-shot pruning.** Phase I performs $G$ saliency computations (one per tier), each a single backward pass on $\mathcal{D}_{\text{valid}}$ with per-layer aggregation and cross-layer consistency checks.
**Per-round valuation.** Tier-level Shapley uses Monte Carlo permutations over $G$ tiers; utilities reuse cached inner products to compute equation 7. Complexity is $\mathcal{O}(MG)$ for permutations and $\mathcal{O}(|\mathcal{S}_t|)$ for dot-products; within-tier allocation is up to $\mathcal{O}(\sum_g |P_{g,t}|^2)$ and can be reduced via caps or pair subsampling.
**Privacy.** Neither phase accesses client raw data. Phase I and BN use only public/unlabeled data; valuation depends solely on model deltas $\{\Delta \boldsymbol{w}_{i,t}\}$.

## 5 Experiments

### 5.1 Benchmarks and data partitions

We evaluate on CIFAR-10/100 Krizhevsky (2009), FEMNIST and Shakespeare (LEAF) Caldas et al. (2018). For CIFAR-10/100 we create $N$=100 clients via Dirichlet sampling with $\alpha \in \{0.1, 0.5\}$ Hsu et al. (2019). FEMNIST uses per-writer user partitions; Shakespeare uses speaker partitions. Vision metrics are top-1 accuracy; language metrics are character accuracy and perplexity. All methods share identical training schedules and sampling.

### 5.2 Models, exits, and tiers

**Backbones and exits.** Vision uses *ResNet-110* He et al. (2016) with four exits (after conv2_x, conv3_x, conv4_x, and the final head). Shakespeare uses a 4-layer GRU Cho et al. (2014) with exits after each layer; the multi-exit objective in Eq. 2 is used throughout.

**Compute schedule.** We fix a geometric per-tier budget: base FLOPs $F_1$ (full model) and $F_g = \rho^{g-1} F_1$ with $\rho = 0.5$ for $g = 2, 3, 4$. In SNOWFL Phase I we choose $(d_g, \boldsymbol{c}_g)$ to satisfy FLOPs$(\boldsymbol{c}_g, d_g) \leq F_g$; masks are frozen. Baselines are compute-matched to the same tier budgets (within $\pm 2\%$).

**Baselines** We compare against HeteroFL Diao et al. (2020), DepthFL Kim et al. (2022), ScaleFL Ilhan et al. (2023), InclusiveFL Liu et al. (2022a), and ReeFL Lee et al. (2024), each tuned to match $F_g$.

### 5.3 Main results

Table 1 reports test accuracy (best-exit/exit-all, compute-matched) on CIFAR-10/100 at $\alpha \in \{0.1, 0.5\}$ and FEMNIST and Shakespeare. SNOWFL consistently outperforms baselines. On CIFAR-10, it is ahead by +3.26 points against the next best at $\alpha$=0.5 and by +9.05 points at $\alpha$=0.1. On CIFAR-100, it reaches 41.0% at $\alpha$=0.5 (best baseline: 36.0%), and is slightly ahead at $\alpha$=0.1. SNOWFL is best on both FEMNIST and Shakespeare, with a +3.0 point gap on Shakespeare. Overall, the best rational relative improvement is **15%**.

---

**Algorithm 1** SNOWFL: One-shot SNIP pruning with data-free Owen valuation and capacity-constrained reassignment

---

**Require:** Base model $f(\cdot; \boldsymbol{w})$ with exits $\{d_b\}_{b=1}^B$; public/unlabeled set $\mathcal{D}_{\text{valid}}$; tiers $g = 1..G$; compute ratio $\rho \in (0,1)$; capacities $\mathbf{C} = (C_1, \ldots, C_G)$; rounds $T$; local steps $K$; permutations $M$; reassignment period $T_{\text{reg}}$; warm-up $T_{\text{warm}}$; mixture weights $\{\gamma_t, \alpha_t\}$.

1: **Phase I (one shot): Saliency-guided tier masks**
2: Compute SNIP saliencies on $\mathcal{D}_{\text{valid}}$ (Eq. 4); set $F_g = \rho^{g-1} F_1$.
3: For each $g$: aggregate to unit-level scores; choose deepest feasible exit $d_g$ and per-layer TopK $\{\kappa_g^{(l)}\}$ s.t. FLOPs$(\boldsymbol{c}_g, d_g) \leq F_g$; enforce cross-layer consistency; freeze $\boldsymbol{c}_g$.

4: **Phase II (federation across heterogeneous tiers)**
5: **for** $t = 1$ **to** $T$ **do**
6:     Sample clients $\mathcal{S}_t$; send masked models $(\boldsymbol{w}_t \odot \boldsymbol{c}_{g(i,t)})$.
7:     **for** $i \in \mathcal{S}_t$ **do**
8:         Train $K$ local steps with the multi-exit loss (Eq. 2) restricted to tier $g(i,t)$; return $\Delta \boldsymbol{w}_{i,t}$ (zeros on pruned coords).
9:     **end for**
10:     Compute normalized target direction $\boldsymbol{v}_t$ (Eq. 6) and tier partition $\mathcal{P}_t = \{P_{g,t}\}$.
11:     *Tier valuation (quotient game):* with $U_t(A) = \sum_{i \in A} [\langle \Delta \boldsymbol{w}_{i,t}, \boldsymbol{v}_t \rangle]_+^{1/2}$, estimate $\{\phi_{g,t}\}$ via $M$ random permutations (Eq. 8); clamp negatives; normalize $\sum_g \phi_{g,t} = U_t^{\text{grp}}(\{1..G\})$.
12:     **for** each $g$ with $P_{g,t} \neq \varnothing$ **do**
13:         For $i \in P_{g,t}$: compute alignment $a_{i,t} = [\langle \Delta \boldsymbol{w}_{i,t}, \boldsymbol{v}_t \rangle]_+$ and peer-diversity $d_{i,t}$ (Eq. 9).
14:         Combine (Eq. 10) to $z_{i,t}$; set $w_{i,t} = \exp(z_{i,t})/\sum_{j \in P_{g,t}} \exp(z_{j,t})$; allocate $v_{i,t} = \phi_{g,t} w_{i,t}$.
15:     **end for**
16:     Size weights: $\tilde{\alpha}_{i,t} = n_i / \sum_{j \in \mathcal{S}_t} n_j$.
17:     *Contribution-weighted aggregation:* $\boldsymbol{w}_{t+1} = \boldsymbol{w}_t + \eta_t \sum_{i \in \mathcal{S}_t} \beta_{i,t} \Delta \boldsymbol{w}_{i,t}$,    $\beta_{i,t} \propto v_{i,t} \tilde{\alpha}_{i,t}$, $\sum_i \beta_{i,t} = 1$.
18:     *BN stabilization:* refresh BN stats on $\mathcal{D}_{\text{valid}}$ or via a brief unlabeled client pass.
19:     **if** $t > T_{\text{warm}}$ **and** $t \bmod T_{\text{reg}} = 0$ **then**
20:         Rank clients by recent $v$ (carry-forward for non-participants); reassign into $G$ tiers under capacities $\mathbf{C}$; reuse frozen $(\boldsymbol{c}_g, d_g)$.
21:     **end if**
22: **end for**

---

Table 1: Mean test accuracy (%) on CIFAR-10/100, FEMNIST, and Shakespeare. Best in **bold**.

| Method | CIFAR-10 | | CIFAR-100 | | FEMNIST | Shakespeare |
|---|---|---|---|---|---|---|
| | $\alpha$=0.1 | $\alpha$=0.5 | $\alpha$=0.1 | $\alpha$=0.5 | | |
| HeteroFL Diao et al. (2020) | 31.74 | 67.58 | 20.80 | 32.96 | 80.42 | 51.0 |
| DepthFL Kim et al. (2022) | 33.59 | 69.49 | 24.89 | 36.04 | 83.18 | 52.0 |
| InclusiveFL Liu et al. (2022a) | 29.26 | 70.97 | 23.38 | 34.94 | 82.91 | 52.8 |
| ReeFL Lee et al. (2024) | 32.70 | 70.37 | 23.78 | 35.20 | 84.20 | 52.4 |
| ScaleFL Ilhan et al. (2023) | 36.88 | 71.58 | 23.63 | 34.53 | 83.30 | 52.3 |
| **SNOWFL (ours)** | **45.93** | **74.84** | **24.95** | **41.00** | **84.22** | **55.4** |

## 5.4 Per-tier budgets and system details

**Per-tier compute.** Table 2 reports FLOPs (per forward, MMac) and parameters (K) at each exit of ResNet-110 for two families of baselines. InclusiveFL, DepthFL and ReeFL use the canonical early-exit computation schedule; ScaleFL, HeteroFL and SNOWFL share a

Table 2: Per-tier compute for ResNet-110. Comparison of FLOPs (MMac) and parameters (K) at designated network exits for three groups of methods.

| Tier | InclusiveFL / DepthFL / ReeFL | | ScaleFL / SNOWFL | | HeteroFL | |
|---|---|---|---|---|---|---|
| | FLOPs | Params (K) | FLOPs | Params (K) | FLOPs | Params (K) |
| 1 | 98.75 | 130.23 | 99.20 | 132.00 | 97.70 | 129.20 |
| 2 | 163.36 | 384.09 | 158.00 | 360.00 | 159.10 | 368.00 |
| 3 | 207.63 | 800.06 | 201.40 | 760.00 | 203.20 | 770.00 |
| 4 | 286.00 | 2030.00 | 286.00 | 2030.00 | 286.00 | 2030.00 |

slightly lighter schedule at exits 2–3 with the same final budget. These are the budgets used to compute-match all methods.

**Further results.** Detailed ablations, sensitivity and convergence panels appear in Appendix C.

## 6 CONCLUSION

We presented SNOWFL, a federated learning framework that combines one-shot, server-side SNIP pruning, a fixed ladder of early exits, and Owen-style contribution weighting. The result is a single global model that different devices can train at different depths and widths, while still aggregating cleanly. A brief BN calibration keeps statistics consistent across pruned paths. Together, these pieces lower compute and communication without adding heavy coordination. Under matched FLOPs/parameter budgets, SNOWFL attains the highest accuracy across all datasets and heterogeneity levels. On CIFAR-10 it leads at both $\alpha$=0.5 and $\alpha$=0.1, and on CIFAR-100 it reaches 41% at $\alpha$=0.5 and 36% at $\alpha$=0.1. In the most heterogeneous scenario, SNOWFL achieves a *relative* accuracy gain of up to 15% over the next best method. Ablations show Owen contribute has a slightly larger standalone effect, while using them together yields the strongest results. On the theory side, we prove a non-convex convergence-to-criticality rate and, under strong convexity, linear convergence to a neighborhood. The bounds account for per-coordinate coverage, grouped heterogeneity, local-step drift, staleness, and early-exit effects, adapting standard smoothness arguments to masked, multi-exit training.

**Limitations and future work.** Our calibration uses public or unlabeled data that may be imperfect; differentially private or synthetic options are worth exploring. The current masks are one-shot and tier-structured; hardware-aware structured sparsity could yield further speedups. Learned tier/exit policies and privacy-preserving contribution estimators are promising directions for improving stability, fairness, and robustness. Overall, SNOWFL offers a simple, low-overhead path to training heterogeneous subnetworks with principled client weighting, delivering a strong accuracy–efficiency trade-off for cross-device FL.

**Reproducibility Statement.** An anonymized repository is available at `https://anonymous.4open.science/r/snowfl-648D/`. The `src/` directory contains reference implementations of Owen-based valuation, SNIP pruning, early-exit models, server strategies, and client code; the provided bash scripts reproduce our runs without additional configuration (Python 3.9, PyTorch 2.3 with CUDA 12.4, NVIDIA RTX 3070 8 GB, driver 550.163.01).

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

## A CONVERGENCE OF GROUPING (OWEN) TO SHAPLEY UNDER ITERATIVE REGROUPING

**Plain-language summary.** As we keep splitting client groups into smaller ones, the set of "legal" permutations used by the Owen value grows. Once groups are singletons, those permutations are all permutations, so the Owen value matches the Shapley value.

We show that the *grouping-only* (Owen-style) valuation used in Section 4.3 converges to the Shapley value under mild, checkable conditions on per-round utilities and the regrouping schedule. The result is independent of any *Owen sampling* (multilinear-extension) estimators; it relies purely on the coalition-structure (grouping) viewpoint.

**Standing notation.** Let $N = \{1, \ldots, |N|\}$ be the client set and $\nu : 2^N \to \mathbb{R}$ a coalition utility (e.g., one induced by equation 7). The Shapley value (per round or for a fixed game) is

$$\phi_i(\nu) \;=\; \frac{1}{|N|!} \sum_{\pi \in \Pi(N)} \big[ \nu(P_i^\pi \cup \{i\}) - \nu(P_i^\pi) \big], \tag{12}$$

with $\Pi(N)$ the set of all permutations of $N$ and $P_i^\pi$ the set of predecessors of $i$ in $\pi$ Shapley (1953). A *partition* (coalition structure) $\mathcal{P} = \{P_1, \ldots, P_m\}$ induces the set $\Pi(\mathcal{P})$ of *compatible permutations*: each block $P_g$ appears contiguously, but the blocks themselves are

permuted arbitrarily, and clients inside each block are internally permuted Owen (1977). The Owen value of player $i \in P_g$ is the compatible-permutation average

$$\text{Ow}_i(\nu, \mathcal{P}) \; = \; \frac{1}{|\Pi(\mathcal{P})|} \sum_{\rho \in \Pi(\mathcal{P})} \big[ \nu(P_i^\rho \cup \{i\}) - \nu(P_i^\rho) \big]. \tag{13}$$

See Owen (1977). *Intuition.* $P_i^\pi$ are simply the clients that appear before $i$ in the ordering $\pi$; the bracketed term is $i$'s marginal gain when added after its predecessors.

## A.1 Assumptions and definitions

**Assumption A.1** (Bounded marginals)**.** There exists $B < \infty$ such that for all $i \in N$ and $S \subseteq N \setminus \{i\}$, $\big| \nu(S \cup \{i\}) - \nu(S) \big| \leq B$.

*Comment.* With equation 7, $\boldsymbol{v}_t$ is unit length and each $\Delta \boldsymbol{w}_{i,t}$ is bounded in norm (via clipping or step-size caps). Then every marginal gain is bounded, so a uniform constant $B$ exists and the assumption holds.

**Definition A.2** (Refinement)**.** A partition $\mathcal{P}' = \{P_1', \ldots, P_{m'}'\}$ is a *refinement* of $\mathcal{P} = \{P_1, \ldots, P_m\}$, written $\mathcal{P}' \succeq \mathcal{P}$, if every $P_j'$ is contained in some $P_g$. Equivalently, $\mathcal{P}'$ is obtained from $\mathcal{P}$ by splitting blocks (no cross-block merges) Owen (1977).

**Assumption A.3** (Eventual refinement)**.** Let $\{\mathcal{P}_t\}_{t \geq 0}$ denote the round-$t$ partition used by Section 4.3. There exists (possibly random) $T$ such that for all $t \geq T$, $\mathcal{P}_{t+1} \succeq \mathcal{P}_t$, and the process almost surely reaches the *finest partition* $\mathcal{P}_{\text{fine}} = \big\{ \{i\} \; : \; i \in N \big\}$ in finite time or as $t \to \infty$.

*Comment.* In our implementation, reassignment respects capacity and only *splits* tiers by valuation (no merging previously separated clients), which operationalizes Assumption A.3. If occasional merges are allowed, the theorem below still holds along any subsequence of pure refinements; see Remark A.9.
*In words.* After some time, we only split existing groups and never merge them, and we eventually end up with groups of size one.

## A.2 Two basic lemmas

**Lemma A.4** (Permutation support under refinement)**.** *If $\mathcal{P}' \succeq \mathcal{P}$ then $\Pi(\mathcal{P}) \subseteq \Pi(\mathcal{P}') \subseteq \Pi(N)$ and*

$$|\Pi(\mathcal{P})| \; = \; m! \prod_{g=1}^{m} |P_g|!, \quad |\Pi(\mathcal{P}')| \; = \; m'! \prod_{j=1}^{m'} |P_j'|!. \tag{14}$$

*In particular, splitting a block strictly increases $|\Pi(\cdot)|$ and, at singletons, $|\Pi(\mathcal{P}_{\text{fine}})| = |N|!$.*

*Proof.* Every ordering that is compatible with $\mathcal{P}$ remains compatible after splitting its blocks, hence $\Pi(\mathcal{P}) \subseteq \Pi(\mathcal{P}')$. Counting equation 14 follows from permuting $m$ blocks and, inside each block, permuting its members Owen (1977). $\square$

**Lemma A.5** (A TV-distance bias bound)**.** *Let $U$ be the uniform distribution on $\Pi(N)$ and $U_\mathcal{P}$ the uniform distribution on $\Pi(\mathcal{P})$. For any function $f$ bounded by $\|f\|_\infty \leq B$,*

$$\Big| \mathbb{E}_U[f] - \mathbb{E}_{U_\mathcal{P}}[f] \Big| \; \leq \; B \cdot \text{TV}\big(U, U_\mathcal{P}\big) \; = \; B\Big(1 - \frac{|\Pi(\mathcal{P})|}{|N|!}\Big),$$

*where* TV *is total-variation distance.*[1]

*Meaning.* Averaging over a subset of permutations (Owen) instead of all permutations (Shapley) introduces at most a $B$-scaled bias that shrinks as the subset approaches the full set.

---

[1]Standard inequality: for probability measures $P, Q$ on a common space, $|\mathbb{E}_P f - \mathbb{E}_Q f| \leq \|f\|_\infty \text{TV}(P, Q)$. Any probability text suffices.

### A.3 Main convergence theorem

**Idea in one line.** Refinement increases the set of compatible permutations until it equals all permutations; the corresponding averages of the same bounded marginal function must converge.

**Theorem A.6** (Owen $(\mathcal{P}_t) \to$ Shapley under refinement). *Under Assumptions A.1 and A.3, for every $i \in N$,*

$$\lim_{t\to\infty} \mathbb{E}\Big[\mathrm{Ow}_i(\nu,\mathcal{P}_t)\Big] = \phi_i(\nu).$$

*Moreover, at any $t$,*

$$\left| \mathbb{E}\big[\mathrm{Ow}_i(\nu,\mathcal{P}_t)\big] - \phi_i(\nu) \right| \leq B\left(1 - \frac{|\Pi(\mathcal{P}_t)|}{|N|!}\right) = B\left(1 - \frac{m_t! \prod_{g=1}^{m_t} |P_{g,t}|!}{|N|!}\right), \qquad (15)$$

*where $\mathcal{P}_t = \{P_{1,t}, \ldots, P_{m_t,t}\}$.*

*Proof.* By equation 13 and equation 12, both values are uniform averages of the same bounded marginal function over two permutation sets. Lemma A.5 gives the bias bound equation 15. Lemma A.4 plus Assumption A.3 imply $|\Pi(\mathcal{P}_t)| \uparrow |N|!$, so the bound goes to zero and the Owen value converges to the Shapley value. $\qquad\square$

**Corollary A.7** (Singletons). *If $\mathcal{P}_t = \mathcal{P}_{\mathrm{fine}}$ for some $t$, then $\mathrm{Ow}_i(\nu,\mathcal{P}_t) = \phi_i(\nu)$ for all $i$.*

*Remark* A.8 (Averaging across coalition structures). There is a complementary, classical identity: the Shapley value equals a suitable *average* of Owen values over coalition structures with the same block-size multiset López & Saboya (2009). Thus, even without reaching singletons, a policy whose long-run distribution over partitions matches those weights yields time-average Owen $\to$ Shapley. We cite this only for context; our proof does not rely on it.

*Remark* A.9 (Subsequence of refinements). If reassignment occasionally merges across previously split blocks, consider the subsequence of *refinement times* where $\mathcal{P}_{t_{k+1}} \succeq \mathcal{P}_{t_k}$. The bound equation 15 and the same limit apply along $\{t_k\}$ as soon as $\mathcal{P}_{t_k} \to \mathcal{P}_{\mathrm{fine}}$.

### A.4 Link to our per-round implementation

**Utilities.** With standard training controls (clipping or LR caps), equation 7 yields bounded nonnegative marginals, so Assumption A.1 holds.

**Quotient game and intra-block split.** The tier-level Shapley on the quotient game, followed by an efficiency-preserving split within each tier, is exactly the Owen two-step.

**Refinement.** Our capacity-constrained reassignment can be run in "split-only" mode, satisfying Assumption A.3; if merges occur, the convergence still holds along any refinement subsequence (Remark A.9).

## B Convergence of SNOWFL

**Plain-language summary.** The server updates using a lightly noisy averaged gradient built from masked local steps. We bound that noise in terms of group similarity, coverage of each coordinate, staleness, masking, and local drift. Summing the descent shows the gradient norms average to a small value (non-convex case), and under strong convexity we contract linearly to a fixed neighborhood.

We analyze the global update

$$\boldsymbol{w}^{t+1} = \boldsymbol{w}^t - \eta \widehat{\boldsymbol{g}}^t, \qquad \eta > 0, \qquad (16)$$

where $\widehat{\boldsymbol{g}}^t$ is a coordinate-wise normalized, group-aware gradient estimator. Each client runs $E$ local masked-SGD steps with stepsize $\gamma$ starting from $\boldsymbol{w}^t$. Let $\mathcal{S}_t$ be the participating set at round $t$, and let $P_{g,t}$ denote the participants in group $g$. Let $\boldsymbol{P}_g \in \{0,1\}^{d \times d}$ denote

the diagonal mask for group $g$ (retained coordinates are ones). For coordinate $j$, define the per-coordinate participation count

$$\Gamma_t^{(j)} := \sum_{i \in \mathcal{S}_t} \mathbb{I}\big\{(\boldsymbol{P}_{g(i,t)})_{jj} = 1\big\},$$

and the normalized per-coordinate weights

$$\alpha_{i,t}^{(j)} := \frac{\beta_{i,t}(\boldsymbol{P}_{g(i,t)})_{jj}}{\sum_{k \in \mathcal{S}_t} \beta_{k,t}(\boldsymbol{P}_{g(k,t)})_{jj}}, \qquad \alpha_{i,t}^{(j)} \geq 0, \ \sum_{i \in \mathcal{S}_t} \alpha_{i,t}^{(j)} = 1, \qquad (17)$$

for nonnegative aggregation weights $\beta_{i,t}$ with $\sum_{i \in \mathcal{S}_t} \beta_{i,t} = 1$. *Well-definedness.* By coverage (Assumption A5), $\Gamma_t^{(j)} \geq 1$ for all active $j$, so the denominator in equation 17 is nonzero.

The server estimator and local iterates are

$$(\widehat{\boldsymbol{g}}^t)_j = \sum_{i \in \mathcal{S}_t} \alpha_{i,t}^{(j)} (\bar{\boldsymbol{g}}_i^t)_j, \qquad \bar{\boldsymbol{g}}_i^t = -\frac{\boldsymbol{w}_i^{t,E} - \boldsymbol{w}^t}{\gamma E}, \qquad \boldsymbol{w}_i^{t,e+1} = \boldsymbol{w}_i^{t,e} - \gamma \boldsymbol{P}_{g(i,t)} \widetilde{\boldsymbol{g}}_i^{t,e}, \quad (18)$$

for $e = 0, \ldots, E-1$ with $\boldsymbol{w}_i^{t,0} = \boldsymbol{w}^t$. We write $\boldsymbol{g}(\boldsymbol{w}) = \nabla F(\boldsymbol{w})$, $\boldsymbol{g}_i(\boldsymbol{w}) = \nabla F_i(\boldsymbol{w})$, and $\boldsymbol{g}_{g,t}(\boldsymbol{w}) = |P_{g,t}|^{-1} \sum_{i \in P_{g,t}} \boldsymbol{g}_i(\boldsymbol{w})$.

**Weights and conditioning.** Let $p_i$ be the (round-$t$) client weight (uniform over current participants by default), $P_{g,t} = \sum_{i \in P_{g,t}} p_i$ the group mass, $\boldsymbol{g}(\boldsymbol{w}) = \sum_i p_i \boldsymbol{g}_i(\boldsymbol{w})$, and $\boldsymbol{g}_{g,t}(\boldsymbol{w}) = P_{g,t}^{-1} \sum_{i \in P_{g,t}} p_i \boldsymbol{g}_i(\boldsymbol{w})$. We write $\mathbb{E}_i[\cdot]$ for expectation w.r.t. $p_i$ and $\mathbb{E}_g[\cdot]$ w.r.t. $P_{g,t}$. All expectations below are taken *conditional on the current partition* $\mathcal{P}_t$.

Throughout, define

$$e^t := \widehat{\boldsymbol{g}}^t - \boldsymbol{g}(\boldsymbol{w}^t), \qquad \bar{\sigma}^2 := \frac{1}{N} \sum_{i=1}^N \sigma_i^2, \qquad \pi_{g,t} := \sum_{i \in P_{g,t} \cap \mathcal{S}_t} \beta_{i,t}, \qquad \bar{\pi}_g := \frac{1}{T} \sum_{t=0}^{T-1} \pi_{g,t},$$

and the average squared step $\overline{\Delta w^2} := \frac{1}{T} \sum_{t=0}^{T-1} \mathbb{E}\|\boldsymbol{w}^{t+1} - \boldsymbol{w}^t\|_2^2$.

**Roadmap.** (i) *Decomposition:* Lemma B.1 splits global gradient variance into within– and across–group parts. (ii) *Local drift:* Lemma B.2 bounds the bias from $E$ masked local steps. (iii) *Masked aggregation:* Lemma B.3 controls $e^t$ via coverage $\Gamma_{\min}$, per–coordinate balancing $c_w$, staleness $K$, and masking noise $\delta$. (iv) *Descent and summation:* the one–step inequality follows from smoothness and Young's inequality, then summation yields the non-convex rate (Theorem B.4), a similarity refinement (Corollary B.5), and the strongly convex neighborhood (Theorem B.6). Our descent pattern follows standard FL proofs (cf. Wang et al. (2023); Tan et al. (2022)).

**Assumptions used throughout this appendix.** We collect the standing conditions referenced by Lemmas B.1–B.3 and Theorems B.4–B.6. These mirror the main text.

(A1) *L-smoothness.* Each $F_i$ and $F = \sum_i p_i F_i$ is $L$-smooth: $F(\boldsymbol{y}) \leq F(\boldsymbol{x}) + \langle \nabla F(\boldsymbol{x}), \boldsymbol{y} - \boldsymbol{x} \rangle + \frac{L}{2}\|\boldsymbol{y} - \boldsymbol{x}\|_2^2$.

(A2) **Unbiased stochastic gradients with bounded variance.** For minibatch $\xi$, $\mathbb{E}[\widetilde{\boldsymbol{g}}_i(\boldsymbol{w}; \xi)] = \boldsymbol{g}_i(\boldsymbol{w})$ and $\mathbb{E}\|\widetilde{\boldsymbol{g}}_i(\boldsymbol{w}; \xi) - \boldsymbol{g}_i(\boldsymbol{w})\|_2^2 \leq \sigma_i^2$.

(A3) **Grouped heterogeneity (intra/inter).** For all $\boldsymbol{w}$ and groups $P_{g,t}$,

$$\mathbb{E}_{i \in P_{g,t}}\|\boldsymbol{g}_i(\boldsymbol{w}) - \boldsymbol{g}_{g,t}(\boldsymbol{w})\|_2^2 \leq \sigma_{\text{intra},g}^2, \qquad \mathbb{E}_g\|\boldsymbol{g}_{g,t}(\boldsymbol{w}) - \boldsymbol{g}(\boldsymbol{w})\|_2^2 \leq \sigma_{\text{inter}}^2.$$

(A4) **Gradient norm bound (used in similarity corollary).** $\|\boldsymbol{g}_i(\boldsymbol{w})\|_2 \leq G$ for all $i, \boldsymbol{w}$.

(A5) **Per-coordinate coverage.** For $\Gamma_t^{(j)} = \sum_{i \in \mathcal{S}_t} \mathbb{I}\{(\boldsymbol{P}_{g(i,t)})_{jj} = 1\}$, $\min_{t,j} \Gamma_t^{(j)} \geq \Gamma_{\min} \geq 1$.

(A6) **Per-coordinate balancing cap.** For all active $j$, $\sum_{i \in \mathcal{S}_t}(\alpha_{i,t}^{(j)})^2 \leq c_w/\Gamma_t^{(j)}$. (Uniform per-coordinate averaging satisfies $c_w{=}1$.)

(A7) **Bounded staleness.** Each local gradient is evaluated on an iterate at most $K$ rounds old.

(A8) **Masking/model-reduction error.** For all $i, g, \boldsymbol{w}$, $\mathbb{E}\| \boldsymbol{P}_g \boldsymbol{g}_i(\boldsymbol{w}) - \boldsymbol{g}_i(\boldsymbol{w})\|_2^2 \leq \delta^2$.

(A9) **Early-exit Lipschitzness.** For each exit $b$, $\|f_{\leq d_b}(x; \boldsymbol{w}) - f_{\leq d_b}(x; \boldsymbol{w}')\|_2 \leq L_b \|\boldsymbol{w} - \boldsymbol{w}'\|_2$ for all $(x, \boldsymbol{w}, \boldsymbol{w}')$.

(A10) **Strong convexity (used only in Theorem B.6).** $F$ is $\mu$-strongly convex on the retained subspace.

We use $C_1, C_2, \tilde{C} > 0$ for universal constants that do not depend on $t$ or problem size.

**Assumption usage.** $L$–smoothness: Lemma B.2, Lemma B.3. Bounded variance $\{\sigma_i^2\}$: Lemma B.2, Lemma B.3. Grouped heterogeneity ($\sigma_{\text{intra},g}^2, \sigma_{\text{inter}}^2$): Lemma B.1, Lemma B.3. Coverage $\Gamma_{\min}$ and balancing $c_w$: Lemma B.3. Staleness $K$: Lemma B.3. Masking error $\delta$: Lemma B.3. Early–exit Lipschitz $L_b$: equation 25. Strong convexity $\mu$: Theorem B.6.

### B.1 Variance decomposition and local-step drift

**Intuition (decomposition).** Grouping helps by shrinking within–group dispersion while leaving the dispersion of group means unchanged.

**Lemma B.1** (Variance decomposition (grouped; conditional on $\mathcal{P}_t$)). *For any $\boldsymbol{w}$,*

$$\mathbb{E}_i \|\boldsymbol{g}_i(\boldsymbol{w}) - \boldsymbol{g}(\boldsymbol{w})\|_2^2 = \mathbb{E}_g \mathbb{E}_{i \in P_{g,t}} \|\boldsymbol{g}_i(\boldsymbol{w}) - \boldsymbol{g}_{g,t}(\boldsymbol{w})\|_2^2 + \mathbb{E}_g \|\boldsymbol{g}_{g,t}(\boldsymbol{w}) - \boldsymbol{g}(\boldsymbol{w})\|_2^2. \quad (19)$$

*Proof.* Expand $\boldsymbol{g}_i - \boldsymbol{g} = (\boldsymbol{g}_i - \boldsymbol{g}_{g,t}) + (\boldsymbol{g}_{g,t} - \boldsymbol{g})$ and square; the cross term averages to 0 by centering within each group. $\qquad\square$

**Intuition (local drift).** Masked local SGD deviates from the instantaneous client gradient because of stochastic noise and movement of the iterate during the $E$ steps; smoothness converts iterate motion to gradient mismatch, yielding a $\gamma LE$ scaling.

**Lemma B.2** (Local-step drift after $E$ masked steps). *Under $L$-smoothness and bounded variance,*

$$\mathbb{E}\|\bar{\boldsymbol{g}}_i^t - \boldsymbol{g}_i(\boldsymbol{w}^t)\|_2^2 \leq C_2 \gamma LE (\sigma_i^2 + G^2), \quad (20)$$

*for a universal constant $C_2$.*

*Proof.* Write

$$\bar{\boldsymbol{g}}_i^t = \frac{1}{E} \sum_{e=0}^{E-1} \widetilde{\boldsymbol{g}}_i^{t,e}(\boldsymbol{w}_i^{t,e}) = \boldsymbol{g}_i(\boldsymbol{w}^t) + \frac{1}{E} \sum_{e=0}^{E-1} \left( \widetilde{\boldsymbol{g}}_i^{t,e} - \boldsymbol{g}_i(\boldsymbol{w}_i^{t,e}) \right) + \frac{1}{E} \sum_{e=0}^{E-1} \left( \boldsymbol{g}_i(\boldsymbol{w}_i^{t,e}) - \boldsymbol{g}_i(\boldsymbol{w}^t) \right).$$

Use $\|a + b\|^2 \leq 2\|a\|^2 + 2\|b\|^2$, bounded variance for the first sum, and $L$-smoothness with $\|\boldsymbol{w}_i^{t,e} - \boldsymbol{w}^t\| \leq \sum_{s < e} \gamma \|\boldsymbol{P}_{g(i,t)} \widetilde{\boldsymbol{g}}_i^{t,s}\|$ plus $\mathbb{E}\|\widetilde{\boldsymbol{g}}\|^2 \leq \sigma_i^2 + G^2$ for the second, to obtain equation 20. $\qquad\square$

### B.2 Masked aggregation error (coordinate-wise chain)

**Intuition (masked aggregation).** Per–coordinate normalization averages only across clients that retained that coordinate. Coverage $\Gamma_{\min}$ and the balancing cap $c_w$ yield a $1/\Gamma_{\min}$ improvement; masking noise $\delta$ and staleness $K$ appear additively in second moment.

**Lemma B.3** (Masked aggregation error). *Let $e^t = \widehat{g}^t - g(w^t)$. Under coverage $\Gamma_{\min}$ and the per-coordinate cap $\sum_i (\alpha_{i,t}^{(j)})^2 \leq c_w / \Gamma_t^{(j)}$, staleness $K$, and masking error $\delta$,*

$$\mathbb{E}\|e^t\|_2^2 \leq \underbrace{\frac{c_w}{\Gamma_{\min}} \sum_g \pi_{g,t}\, \sigma_{\mathrm{intra},g}^2 + \sigma_{\mathrm{inter}}^2}_{group\ heterogeneity} + \underbrace{C_1 K^2 \bar{\sigma}^2}_{staleness} + \underbrace{\delta^2}_{masking} + \underbrace{C_2\, \gamma LE\,(\bar{\sigma}^2 + G^2)}_{local\ drift}. \tag{21}$$

*Step-by-step chain.* For coordinate $j$,

$$e_j^t = \sum_i \alpha_{i,t}^{(j)} (\bar{g}_i^t)_j - g_j(w^t)$$

$$= \underbrace{\sum_i \alpha_{i,t}^{(j)} \Big[(\bar{g}_i^t)_j - (g_i(w^t))_j\Big]}_{=:a_j} + \underbrace{\sum_i \alpha_{i,t}^{(j)} \Big[(g_i(w^t))_j - (g(w^t))_j\Big]}_{=:b_j}.$$

Then $\mathbb{E}e_j^{t2} \leq 2\mathbb{E}a_j^2 + 2\mathbb{E}b_j^2$. For $a_j$, Lemma B.2 and the cap give $\mathbb{E}a_j^2 \leq \frac{c_w}{\Gamma_t^{(j)}} C_2 \gamma LE(\bar{\sigma}^2 + G^2)$. For $b_j$, decompose $(g_i - g) = (g_i - g_{g,t}) + (g_{g,t} - g)$, use Lemma B.1 and the same cap:

$$\mathbb{E}b_j^2 \leq \frac{c_w}{\Gamma_t^{(j)}} \sum_g \pi_{g,t}\, \sigma_{\mathrm{intra},g}^2 + \sigma_{\mathrm{inter}}^2.$$

$K$-delayed evaluations add $C_1 K^2 \bar{\sigma}^2$ (iterate drift under smoothness and bounded noise). Masking contributes $\delta^2$ by assumption. Sum over $j$, and use $\Gamma_t^{(j)} \geq \Gamma_{\min}$ to obtain equation 21. $\qquad\square$

### B.3 One-step descent inequality (full chain)

**Intuition (descent).** Write the server step as true gradient plus error; Young's inequality trades a quarter of descent for bounded error growth; choosing $\eta L \leq 1/2$ controls the quadratic term.

By $L$-smoothness (cf. Wang et al. (2023)), for $U_1 := \mathbb{E}\langle g(w^t), w^{t+1} - w^t\rangle$ and $U_2 := \frac{L}{2}\mathbb{E}\|w^{t+1} - w^t\|_2^2$,

$$\mathbb{E}[F(w^{t+1}) - F(w^t)] \leq U_1 + U_2 + E_{\mathrm{exit}}^t. \tag{22}$$

**Displayed chain for equation 22.**

$$U_1 = \mathbb{E}\langle g(w^t), -\eta\,(g(w^t) + e^t)\rangle$$

$$= -\eta\, \mathbb{E}\|g(w^t)\|_2^2 - \eta\, \mathbb{E}\langle g(w^t), e^t\rangle$$

$$\leq -\eta\, \mathbb{E}\|g(w^t)\|_2^2 + \eta\big(\tfrac{1}{4}\mathbb{E}\|g(w^t)\|_2^2 + \mathbb{E}\|e^t\|_2^2\big) \quad \text{(Young)}$$

$$= -\tfrac{3\eta}{4}\, \mathbb{E}\|g(w^t)\|_2^2 + \eta\, \mathbb{E}\|e^t\|_2^2, \tag{23}$$

$$U_2 = \tfrac{L}{2}\, \mathbb{E}\| -\eta(g(w^t) + e^t)\|_2^2$$

$$= \tfrac{L\eta^2}{2}\, \mathbb{E}\|g(w^t)\|_2^2 + L\eta^2\, \mathbb{E}\langle g(w^t), e^t\rangle + \tfrac{L\eta^2}{2}\, \mathbb{E}\|e^t\|_2^2$$

$$\leq \tfrac{L\eta^2}{2}\, \mathbb{E}\|g(w^t)\|_2^2 + L\eta^2\big(\tfrac{1}{4}\mathbb{E}\|g(w^t)\|_2^2 + \mathbb{E}\|e^t\|_2^2\big) + \tfrac{L\eta^2}{2}\, \mathbb{E}\|e^t\|_2^2$$

$$= \tfrac{3L\eta^2}{4}\, \mathbb{E}\|g(w^t)\|_2^2 + 2L\eta^2\, \mathbb{E}\|e^t\|_2^2 \leq \tfrac{\eta}{4}\, \mathbb{E}\|g(w^t)\|_2^2 + \eta\, \mathbb{E}\|e^t\|_2^2, \quad (\eta L \leq \tfrac{1}{2}), \tag{24}$$

and early exits contribute

$$E_{\mathrm{exit}}^t \leq C_{\mathrm{exit}} \sum_{b=1}^B \lambda_b L_b^2\, \mathbb{E}\|w^{t+1} - w^t\|_2^2. \tag{25}$$

Here $\{\lambda_b\}_{b=1}^B$ are the exit-loss weights from the multi-exit objective. Combining equation 22, equation 23, equation 24, equation 25 gives

$$\mathbb{E}[F(w^{t+1}) - F(w^t)] \leq -\tfrac{\eta}{2}\, \mathbb{E}\|g(w^t)\|_2^2 + 2\eta\, \mathbb{E}\|e^t\|_2^2 + E_{\mathrm{exit}}^t. \tag{26}$$

## B.4 Main rates (non-convex, similarity refinement, strongly convex)

**Theorem B.4** (Non-convex rate to criticality). *Fix $\eta \leq 1/(2L)$ and run $T$ rounds. Then*

$$\frac{1}{T}\sum_{t=0}^{T-1}\mathbb{E}\|\boldsymbol{g}(\boldsymbol{w}^t)\|_2^2 \;\leq\; \frac{4\big(F(\boldsymbol{w}^0)-F_\star\big)}{\eta T} \;+\; 8\left[\begin{array}{l}\frac{c_w}{\Gamma_{\min}}\sum_g \bar{\pi}_g\,\sigma_{\mathrm{intra},g}^2 \;+\; \sigma_{\mathrm{inter}}^2\\ +\; C_1 K^2\bar{\sigma}^2 \;+\; \delta^2\\ +\; C_2\,\gamma LE\,(\bar{\sigma}^2+G^2)\end{array}\right] \tag{27}$$

$$+\; \frac{4C_{\mathrm{exit}}}{\eta}\sum_{b=1}^{B}\lambda_b L_b^2\,\overline{\Delta w^2}.$$

*Meaning.* The average squared gradient decays like $1/T$ plus fixed error terms that shrink with better grouping (higher similarity), wider coverage, smaller staleness, milder masking, and shorter/softer local steps.

*Proof.* From equation 26,

$$-\tfrac{\eta}{2}\,\mathbb{E}\|\boldsymbol{g}(\boldsymbol{w}^t)\|_2^2 \;\geq\; \mathbb{E}[F(\boldsymbol{w}^{t+1})-F(\boldsymbol{w}^t)] - 2\eta\,\mathbb{E}\|e^t\|_2^2 - E_{\mathrm{exit}}^t.$$

Sum $t = 0, \ldots, T-1$, telescope, divide by $\eta T$, and multiply by $-2$:

$$\frac{1}{T}\sum_{t=0}^{T-1}\mathbb{E}\|\boldsymbol{g}(\boldsymbol{w}^t)\|_2^2 \;\leq\; \frac{2\big(F(\boldsymbol{w}^0)-F(\boldsymbol{w}^T)\big)}{\eta T} + \frac{4}{T}\sum_t\mathbb{E}\|e^t\|_2^2 + \frac{2}{\eta T}\sum_t E_{\mathrm{exit}}^t.$$

Lower-bound $F(\boldsymbol{w}^T) \geq F_\star$, substitute Lemma B.3, and use equation 25 with the definition of $\overline{\Delta w^2}$ to obtain equation 27. $\qquad\square$

**Constant accounting.** The factor 4 in front of $(F(\boldsymbol{w}^0)-F_\star)/(\eta T)$ and the error average stems from the $-\eta/2$ and $+2\eta$ coefficients in equation 26 after summation and normalization. The aggregated 8 multiplying the heterogeneity-drift block reflects that $e^t$ also enters once inside $U_1$ via Young's inequality (cf. equation 23), which doubles the block upon collecting terms. The exit term picks up $4/\eta$ by the same algebra applied to equation 25.

**Corollary B.5** (Similarity refinement). *Assume $\|\boldsymbol{g}_i(\cdot)\|_2 \leq G$. Define the within–group cosine similarity*

$$\rho_{g,t} \;=\; \frac{2}{|P_{g,t}|(|P_{g,t}|-1)}\sum_{i<j\in P_{g,t}}\frac{\langle\boldsymbol{g}_i,\boldsymbol{g}_j\rangle}{\|\boldsymbol{g}_i\|\,\|\boldsymbol{g}_j\|}, \qquad \bar{\rho}_g \;=\; \frac{1}{T}\sum_{t=0}^{T-1}\mathbb{E}[\rho_{g,t}]. \tag{28}$$

*Then $\sigma_{\mathrm{intra},g}^2 \leq G^2(1-\rho_{g,t})$. Let*

$$\mathcal{E}_{\mathrm{sim}} \;:=\; \frac{c_w G^2}{\Gamma_{\min}}\sum_g \bar{\pi}_g\,(1-\bar{\rho}_g) \;+\; \sigma_{\mathrm{inter}}^2 \;+\; C_1 K^2\bar{\sigma}^2 \;+\; \delta^2 \;+\; C_2\,\gamma LE\,(\bar{\sigma}^2+G^2). \tag{29}$$

*With $\eta \leq 1/(2L)$,*

$$\frac{1}{T}\sum_{t=0}^{T-1}\mathbb{E}\|\boldsymbol{g}(\boldsymbol{w}^t)\|_2^2 \;\leq\; \frac{4\big(F(\boldsymbol{w}^0)-F_\star\big)}{\eta T} \;+\; 8\,\mathcal{E}_{\mathrm{sim}} \;+\; \frac{4C_{\mathrm{exit}}}{\eta}\sum_{b=1}^{B}\lambda_b L_b^2\,\overline{\Delta w^2}. \tag{30}$$

**Theorem B.6** (Strongly convex: linear convergence to a neighborhood). *If $F$ is $\mu$-strongly convex on the retained subspace and $\eta \leq \min\{1/(2L), \mu/(8L^2)\}$, then*

$$\mathbb{E}\|\boldsymbol{w}^{t+1}-\boldsymbol{w}^\star\|_2^2 = \mathbb{E}\|\boldsymbol{w}^t-\boldsymbol{w}^\star-\eta(\boldsymbol{g}(\boldsymbol{w}^t)+e^t)\|_2^2$$

$$= \mathbb{E}\|\boldsymbol{w}^t-\boldsymbol{w}^\star\|_2^2 - 2\eta\,\mathbb{E}\langle\boldsymbol{g}(\boldsymbol{w}^t),\,\boldsymbol{w}^t-\boldsymbol{w}^\star\rangle + \eta^2\,\mathbb{E}\|\boldsymbol{g}(\boldsymbol{w}^t)\|_2^2 + \eta^2\,\mathbb{E}\|e^t\|_2^2$$

$$\leq (1-\eta\mu)\,\mathbb{E}\|\boldsymbol{w}^t-\boldsymbol{w}^\star\|_2^2 + \tfrac{\eta}{4}\,\mathbb{E}\|\boldsymbol{g}(\boldsymbol{w}^t)\|_2^2 + 2\eta^2\,\mathbb{E}\|e^t\|_2^2$$

$$\leq (1-\tfrac{\eta\mu}{2})\,\mathbb{E}\|\boldsymbol{w}^t-\boldsymbol{w}^\star\|_2^2 + \mathcal{R}_t, \tag{31}$$

*where*

$$\mathcal{R}_t := \tilde{C}\left(\eta\,\tfrac{c_w}{\Gamma_{\min}}\sum_g \pi_{g,t}\sigma_{\text{intra},g}^2 + \eta\,\sigma_{\text{inter}}^2 + \eta\,C_1 K^2\bar{\sigma}^2 + \delta^2 + C_2\,\gamma LE\,(\bar{\sigma}^2 + G^2)\right),\quad (32)$$

*for a universal constant $\tilde{C}$.* **Meaning.** *The error contracts by a factor $(1-\eta\mu/2)$ each round, up to a fixed radius set by the same heterogeneity, masking, staleness, and drift terms.*

*Remark* B.7 (Per–coordinate unbiasedness under coverage). If for each coordinate $j$ the active set is a random subset independent of gradients and $\sum_i \alpha_{i,t}^{(j)} = 1$, then $\mathbb{E}[(\hat{\boldsymbol{g}}^t)_j \mid \boldsymbol{w}^t] = g_j(\boldsymbol{w}^t)$. Lemma B.3 then quantifies the residual second moment when actives vary across $j$ and $t$.

*Remark* B.8 (FedAvg as a special case). If $\boldsymbol{P}_g = \boldsymbol{I}$ for all groups (no masking), then $\delta = 0$, $\Gamma_{\min} = |\mathcal{S}_t|$, and $c_w = 1$ under uniform averaging. Bound equation 27 recovers the standard FedAvg nonconvex rate with staleness and local-step drift terms (cf. Tan et al. (2022)).

*Remark* B.9 (One group). With $G{=}1$, we have $\sigma_{\text{inter}}^2 = 0$, $\pi_{1,t} = 1$; the grouped heterogeneity reduces to one intra term, retaining coverage and masking benefits.

**Tuning guide (practical).** (i) *Grouping helps:* increase within-group similarity $\Rightarrow$ lower $\sigma_{\text{intra},g}^2$. (ii) *Coverage matters:* enforce quotas so $\Gamma_{\min}$ stays away from 0. (iii) *Local steps:* keep $\gamma E$ moderate to control local drift. (iv) *Staleness:* smaller $K$ or smaller $\eta$ helps. (v) *Exits:* as training stabilizes, $\overline{\Delta w^2}$ shrinks and the exit penalty vanishes.

**Symbol mini-glossary.** $\Gamma_{\min}$: minimum per-coordinate participation count; $c_w$: per-coordinate balancing cap; $\delta$: masking-model-reduction noise; $K$: staleness bound (delayed gradients); $\bar{\pi}_g$: average group participation weight; $\overline{\Delta w^2}$: average squared update size.

## C  ADDITIONAL EXPERIMENTAL RESULTS

### C.1  ABLATIONS

We ablate **SNIP** (uniform per-layer width), **Owen** (tier-permutation aggregation), and their combination. All settings are compute-matched per tier. Table 3 shows that removing either component consistently hurts across all datasets; dropping *Owen* yields a larger drop than dropping *SNIP*, and removing both is worst. This supports our design choice: *SNIP* stabilizes per-layer capacity while *Owen* aligns tiers during aggregation.

Table 3: Ablation study (mean).

| Configuration | CIFAR-10 (%) | CIFAR-100 (%) | FEMNIST (%) | Shake. (%) |
|---|---|---|---|---|
| Full SNOWFL | 74.8 | 41.0 | 84.2 | 55.4 |
| Without SNIP | 73.6 | 39.3 | 84.1 | 54.2 |
| Without Owen | 72.7 | 38.5 | 83.7 | 53.0 |
| Without Both | 71.2 | 37.0 | 83.5 | 52.1 |

### C.2  CONVERGENCE (ALL METHODS)

Figure 1 reports test accuracy vs. round (best-so-far, lightly smoothed). Methods are close for the first 20 rounds; gaps widen later. SNOWFL consistently reaches the highest accuracy by late rounds, while REEFL is typically the strongest baseline. Gains are most pronounced on CIFAR-100; FEMNIST shows a smaller but persistent edge; Shakespeare saturates smoothly near table values.

### C.3  PER-TIER CONVERGENCE (SNOWFL)

Figure 2 shows exits 0–3 (best-so-far, lightly smoothed). Later exits consistently achieve higher accuracy; the ordering is stable throughout training. The inter-exit gap narrows on

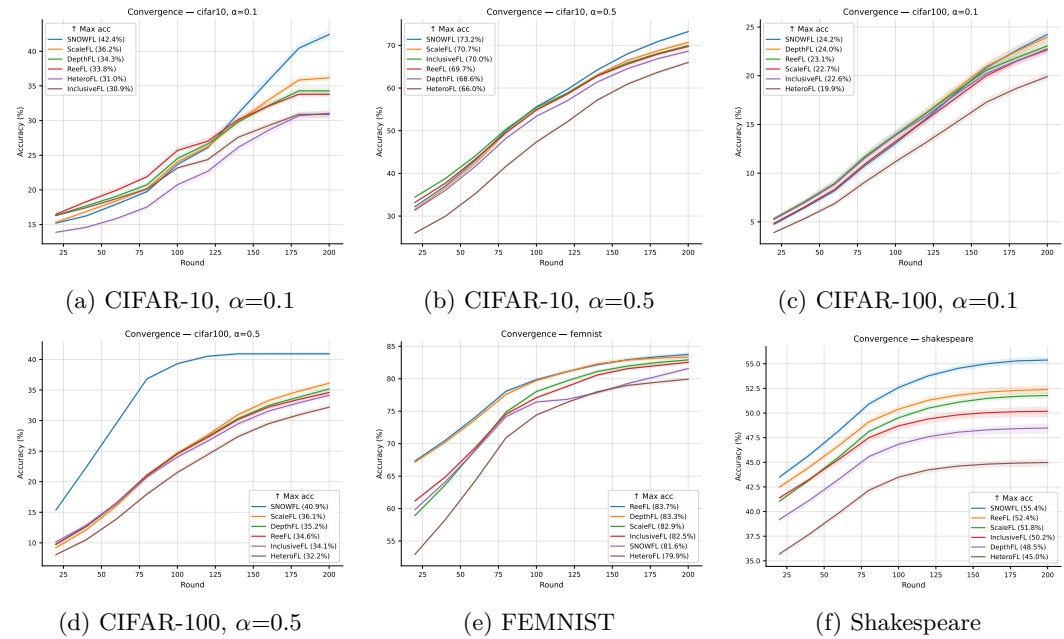

Figure 1: Convergence (all methods). Test accuracy vs. round (best-so-far, lightly smoothed).

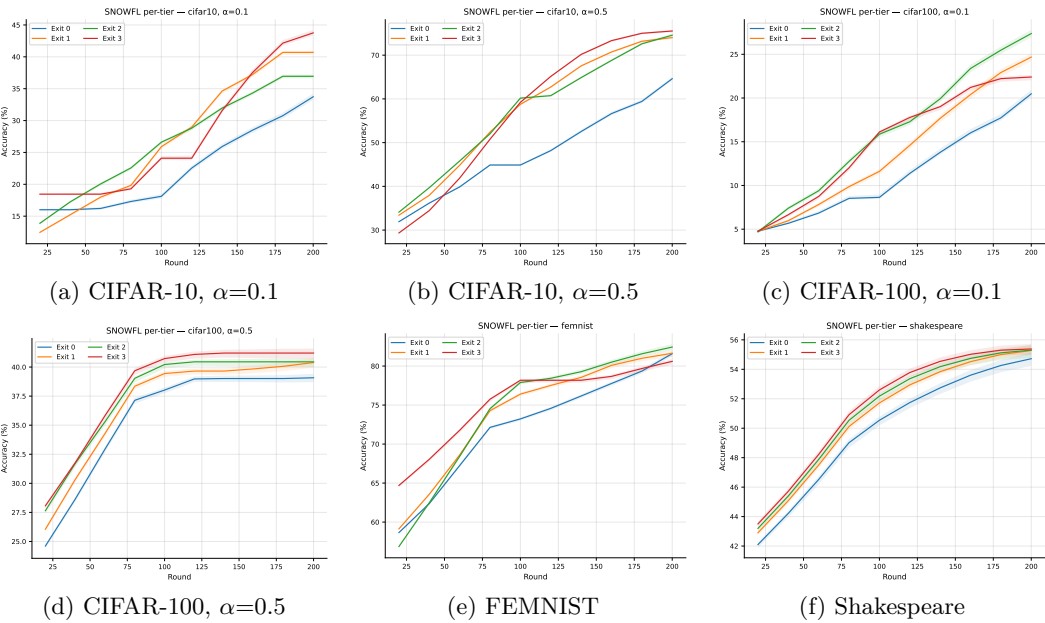

Figure 2: Per-tier convergence for SNOWFL. Exits 0–3 (best-so-far, lightly smoothed).

FEMNIST, while on CIFAR-100 it persists longer. This indicates tiering preserves utility across device classes without sacrificing the strongest exit.

## C.4 SENSITIVITY TO PERMUTATIONS M

Figure 3 sweeps $M$ without seed shadows. A broad optimum occurs around $M \approx 128$ across datasets; gains saturate beyond, with mild degradation at the extremes (very small or very large $M$), suggesting sufficient but not excessive permutation diversity.

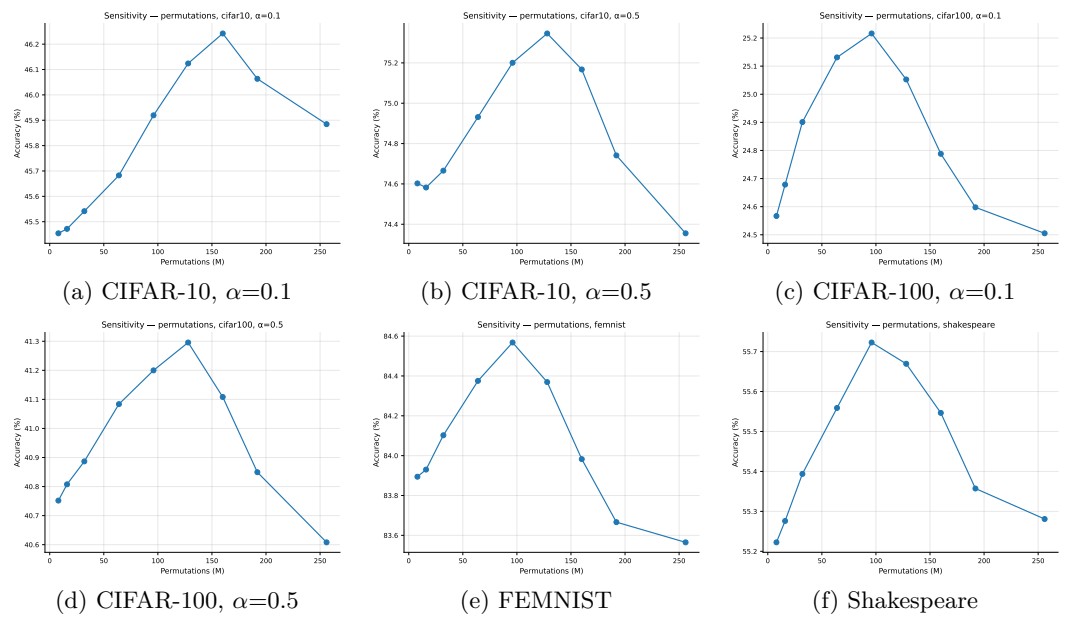

Figure 3: Sensitivity to permutations $M$. Accuracy vs. $M$ (no seed shadow).

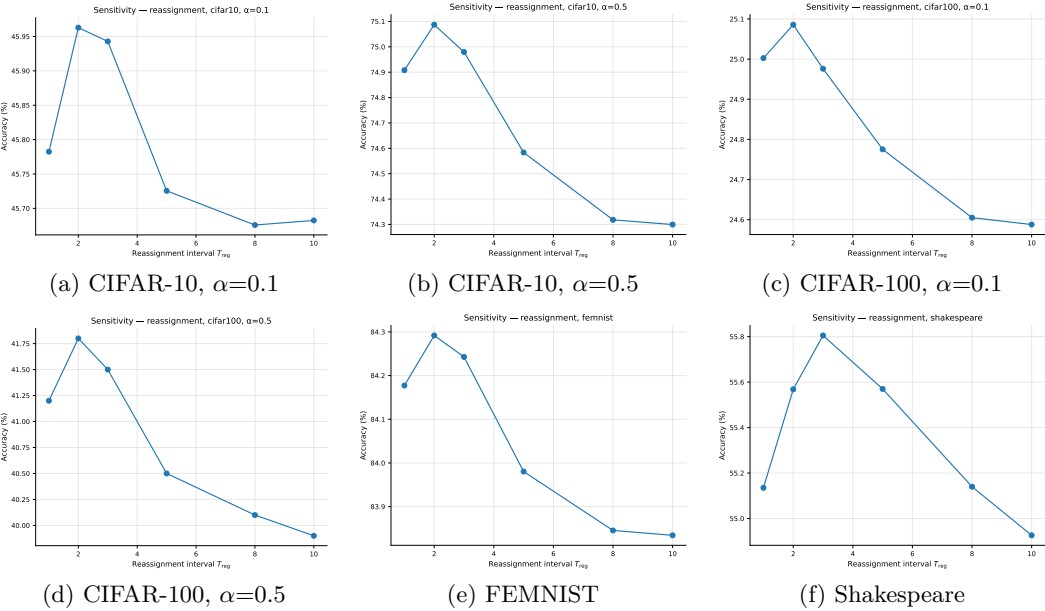

Figure 4: Sensitivity to $T_{\text{reg}}$. Accuracy vs. reassignment interval (no seed shadow).

### C.5 SENSITIVITY TO REASSIGNMENT $T_{\text{reg}}$

Figure 4 sweeps the reassignment cadence. A short interval performs best: $T_{\text{reg}}=2$ tends to be the strongest; $T_{\text{reg}}=1$ is competitive but slightly noisier; performance degrades for $T_{\text{reg}} \geq 5$ as tiers overfit to stale assignments.

