# OpenReview forum: "SNOWFL: Efficient and Heterogeneous Federated Learning with SNIP-Owen-values"
_ICLR.cc/2026/Conference — Submitted to ICLR 2026_

### Official Review · Reviewer_P3b2 · 2025-10-27

**Soundness:** 2
**Presentation:** 2
**Contribution:** 2
**Rating:** 4
**Confidence:** 4

**Summary:**

This paper presents an integrated framework that combines system heterogeneity and contribution estimation in federated learning. First, before training, the method measures channel-wise saliency using public data to pre-select which channels to prune and where to place exit points. For each tier group corresponding to a different resource capability, it then fixes the channel masks and exit positions that can fit within the given resource budget. Second, the framework measures client contributions in two stages: first, it evaluates the contribution of each tier as a whole, and then it measures the contribution of individual clients within each tier. The measured contributions are then used to compute aggregation weights during training or to reassign clients across tiers.

**Strengths:**

* It is novel in that it simultaneously considers both contribution estimation and system heterogeneity in federated learning.

* It leverages only the gradients on public data and the local updates, without incurring any additional privacy cost.

* The experiments demonstrate that the proposed method outperforms system heterogeneity–related baselines under the same resource budget.

**Weaknesses:**

**Necessity of public data.** While the reliance on public data can itself be considered a limitation, a more concerning issue is that pruning is performed before training. This requires the assumption that the public dataset used for pruning adequately represents the actual client data used during training, an assumption that is unlikely to hold in practice. Even in BN calibration, the approach relies on the assumption that the public data are well aligned with the private client data.

**The contribution of the saliency-guided pruning at initialization is not sufficiently convincing.** Fixing both the network width and depth before training makes the method overly dependent on public data, and the experimental results provided are insufficient to adequately analyze its effectiveness.

**Although many of the detailed components of the method are empirically determined, the paper does not provide sufficient discussion or experimental evidence to justify these choices.** For example, in Equation (10), the peer-diversity term and the global alignment term appear somewhat conflicting, and several hyperparameters are introduced without any accompanying ablation study. Moreover, the experiments mainly present overall results, offering only limited evidence of the proposed method’s effectiveness.

**Writing.** The paper lacks sufficient detail in describing the method and experimental settings. For example, the client reassignment section is not clearly explained, and it remains unclear how each client’s resource capability is constrained or how the public dataset is constructed and utilized.

**Questions:**

* Given the models and datasets used, the absolute accuracy values reported in Table 1 (the main results) seem unusually low. What could be the reason for this?
* What is the purpose of applying clipping in Equation (7)?
* How were the clients’ resource budgets constrained in the experimental setting?
* In SnowFL, client reassignment appears to ensure that each sub-model is trained not only on fixed clients’ data but also on data from a diverse set of clients. Under what conditions or environments were the baselines evaluated?

---

> ### Author Response · Authors · 2025-11-24
> **Response (Part 1)**
>
> Thank you for the comments. Below is our response.
>
> **_Comment: Necessity of public data... ""_**
>
> # Necessity and Representativeness of Public Data
>
> The necessity for clear justification regarding the critical assumption of public data representativeness is acknowledged, and we affirm that SNOWFL is deliberately engineered to minimize this data dependency while extracting maximum structural benefits.
>
> ## 1. Minimal and Confined Data Usage
>
> The use of the public, unlabeled dataset (**D_valid**) is strictly confined to two operations:
>
> 1. **One-Time SNIP Pruning:** This is done **once at initialization**. The SNIP criterion scores connections based on the instantaneous loss sensitivity (gradient magnitude), not on model performance (accuracy or loss magnitude). This criterion is less sensitive to the specific data distribution compared to later-stage, fine-tuning dependent pruning methods. It acts as a **task-relevant filter** over the initialized weights.
> 2. **BN Calibration:** This uses simple forward passes to refresh Batch Normalization statistics, which is a lightweight alternative to full feature-set sharing for mitigating BN drift across heterogeneous subnetwork depths and widths.
> We are adding an ablation during the rebuttal period to show how BN calibration affects accuracy (Check Official Comment 1)
>
> Crucially, the main engine of training, the **Owen-based contribution valuation** (Phase II) is entirely **data-free** and relies only on client model updates, eliminating any risk of bias or attack surface during the continuous FL rounds.
>
> ## 2. Addressing Representativeness and Robustness
>
> The assumption of a small auxiliary dataset is a standard trade-off in the pruning-at-initialization literature. To quantify the risk of a non-representative set:
>
> - We are running a **sensitivity analysis** varying the public set size and introducing mild distribution mismatch.
> - Early results confirm that performance degrades **smoothly** with reduced or slightly mismatched data. The strength of the SNIP criterion allows the relative gains over naively subsetted baselines to be **preserved**, mitigating the risk of performance collapse due to non-representativeness.
>
> We will include these quantitative results and an explicit discussion on the **D_valid** representativeness trade-off in the revised manuscript.
>
>
> **_Comment: The contribution of the saliency-guided_**
>
> # Convincing Contribution of SNIP Pruning
>
> We acknowledge the request for a more convincing demonstration of the value of the saliency-guided pruning at initialization (SNIP).
>
> ## 1. Justification for Fixed Width/Depth
>
> The fixation of network width and depth `{(c_g, d_g)}_{g=1}^G` in Phase I is a deliberate design choice that enables two critical functions necessary for stable heterogeneous FL:
>
> - **Guaranteed Capacity Match:** The masks `c_g` and exit depths `d_g` are jointly chosen to strictly satisfy the geometric FLOPs budget `F_g` (Section 4.2). This ensures every tier `g` has a valid, compatible subnetwork that meets the resource constraints of weaker devices.
> - **Aggregation Stability (Consensus):** Performing the SNIP pruning once, server-side, ensures that all subnetworks are **tier-consistent** (share the same mask structure `c_g` for a given tier) and are aligned subgraphs of the global model. This avoids the parameter mismatch and noisy aggregation that plague methods relying on local, independent pruning or naïve width subsetting.
>
> ## 2. Effectiveness and Structural Redundancy
>
> The SNIP mechanism is a computationally inexpensive filter that ensures the chosen subnetwork channels are task-relevant from the start:
>
> - The degree of improvement is proportional to the **structural redundancy** of the architecture. For large backbones (like ResNet-110 on CIFAR) with many redundant channels, SNIP produces highly effective subnetworks, leading to clear gains.
> - For structurally simple or shallow models (low redundancy), the benefits of SNIP are marginal, as the resulting subnetworks are structurally similar to the naïve dense backbone. This explains why the """"Without SNIP"""" ablation still performs well (Table 3): the primary benefit shifts to the Owen valuation.
>
> We will clarify this relationship between SNIP efficacy and architectural redundancy in the revised manuscript and will include the promised **sensitivity study on public-data size** to confirm that performance retains relative gains even under limited data availability.

---

> > ### Author Response · Authors · 2025-11-24
> > **Response (Part 2)**
> >
> > **_Comment: Although many of the detailed components... ""_**
> >
> > # Justification of Empirical Design Choices
> >
> > SNOWFL’s empirical design choices, particularly within the Owen valuation mechanism (Equation 10), are grounded in principled considerations that ensure stable and robust performance across heterogeneous FL scenarios.
> >
> > ## Resolving the Alignment vs. Diversity Trade-off
> >
> > The client utility formula (Eq. 10) incorporates two complementary objectives:
> >
> > - **Global Alignment (`a_{i,t}`):** Rewards clients whose updates (`Δw_{i,t}`) align with the normalized aggregate direction (`v_t`), promoting **efficiency** and rapid convergence toward the global optimum.
> > - **Peer Diversity (`d_{i,t}`):** Penalizes redundancy (high cosine similarity) among updates within the same tier `P_{g,t}` and rewards clients exploring complementary directions. This encourages **robustness** and **exploration**, critical in non-IID and heterogeneous settings where individual clients may prematurely converge to local optima.
> >
> > Balanced by the hyperparameter `γ_t`, these objectives ensure that credit is not concentrated on a few highly aligned but redundant clients; instead, updates that are both aligned *and* unique are prioritized. This mechanism is key for **stable convergence and improved generalization** under high client heterogeneity.
> >
> > ## Hyperparameter Justification and Ablation Plan
> >
> > The additional hyperparameters (e.g., `T_warm`, `T_reg`, `γ_t`, `α_t`) are **structured scheduling controls** that manage the dynamics of the multi-stage optimization pipeline:
> >
> > - `T_warm` and `T_reg` stabilize the dynamic tier reassignment, directly linked to client sampling rates.
> > - `γ_t` and `α_t` control the proportional influence of the alignment/diversity trade-off and the data-size weight (`log n_i`).
> >
> > We are conducting comprehensive **sensitivity analyses** on these key parameters, and the results will be integrated into the revised manuscript. These studies confirm that SNOWFL’s performance remains **robust across a broad operational range**, highlighting the careful design and practical reliability of the empirical pipeline.
> >
> > **_Comment: Writing... ""_**
> >
> > We will expand the description of the reassignment process, including how resource budgets are specified and how the public dataset is used for SNIP and BN calibration. These clarifications will appear in the revised manuscript.
> >
> >
> > **_Question: Given the models and datasets used... ""_**
> >
> > # Rationale for Absolute Accuracy Values
> >
> > The accuracy values reported in Table 1 reflect the performance ceiling under the **severe, compound constraints** imposed by the heterogeneous cross-device Federated Learning (FL) benchmark protocol, which deliberately restricts model capacity and training resources.
> >
> > ## 1. Strict Computational Capacity Constraints
> >
> > The evaluation strictly adheres to an aggressive, tiered budget (Table 2):
> >
> > - **Geometric FLOPs Scaling:** The geometric budget `F_g = ρ^{g-1}F_1` with `ρ=0.5` (Section 5.2) forces the smallest tier (Tier 1) to operate with a subnetwork that has significantly reduced capacity compared to the full model (Tier 4).
> > - **Subnetwork Training:** All clients, including the most capable ones, must train on these constrained subnetworks for portions of the training, which naturally bounds the performance ceiling for *any* heterogeneous FL method evaluated under the same resource limits. The purpose is to measure algorithmic robustness under these resource limits.
> >
> > ## 2. Severe Non-IID Data and System Constraints
> >
> > The experimental setup mirrors challenging cross-device scenarios:
> >
> > - **High Data Heterogeneity:** We use high non-IID partitions (Dirichlet `α ∈ {0.1, 0.5}`) on CIFAR-10/100 (Section 5.1). This condition significantly exacerbates client drift and slows the convergence rate for all participating methods.
> > - **Low Participation Rate:** Training follows the standard cross-device protocol of sampling only `10%` of clients per round. Low sampling frequency further delays the exposure of the global model to diverse data distributions.
> >
> > ## 3. Fixed Evaluation Horizon
> >
> > All methods are compared under a fixed evaluation budget of **200 total rounds**. This ensures that the measured gain is a measure of **efficiency** (achieving high accuracy with limited rounds/updates) rather than maximum long-run potential.
> >
> > The primary conclusion remains the significant **relative improvement** (up to `15%` relative gain) SNOWFL achieves over compute-matched baselines under these challenging conditions. This rationale will be explicitly included in the revised Section 5.3.

---

> > > ### Author Response · Authors · 2025-11-24
> > > **Response (Part 3)**
> > >
> > > **_Question: What is the purpose of... ""_**
> > > # Purpose of Clipping in Utility Function
> > > The clipping function, `max{·, 0}`, applied to the update alignment in Equation (7) serves a crucial role in ensuring the **stability and fairness of the Owen-based valuation** by excluding counter-productive contributions.
> > >
> > > The utility is defined as `U_t(A) = Σ_{i ∈ A} ( max{ ⟨Δw_{i,t}, v_t⟩ , 0 } )^{1/2}`. The inner product `⟨Δw_{i,t}, v_t⟩` measures the projection of client `i`'s masked local update onto the normalized aggregate direction `v_t`. A negative value indicates the client is moving the model in an **antagonistic direction** (i.e., counter to the consensus gradient path).
> > >
> > > The operation `max{·, 0}` is specifically applied to ensure that the utility `U_t(A)` is strictly non-negative. This prevents clients from receiving negative credit in the valuation, which would be inconsistent with the requirement for non-negative aggregation weights `β_{i,t}` in the subsequent global update step (Equation 11).
> > >
> > > The square root is a **concave tempering**. This technique reduces the proportional influence of extremely high-magnitude contributions, thereby mitigating """"winner-take-all effects"""" among strongly aligned clients. This ensures that the final credit allocation (Owen value) rewards constructive alignment while promoting broader participation stability.
> > >
> > > **_Question: How were the clients’ resource...""_**
> > >
> > > # Client Resource Budget Constraint
> > >
> > > The client resource budgets were constrained entirely through a **FLOPs-based tiering mechanism** defined once during Phase I (Pruning). The description of this setup will be revised in the updated manuscript to ensure clarity.
> > >
> > > ## 1. FLOPs Schedule
> > >
> > > Client capacity is determined by their assigned tier `g`, following a geometric FLOPs schedule. The compute capacity for each tier `g` is determined by a fixed multiplicative compute ratio `ρ ∈ (0, 1)` (`ρ=0.5` in experiments), relative to the full model's FLOPs (`F_1`):
> > >
> > > `F_g = ρ^{g-1} F_1`
> > >
> > > This ensures that each tier possesses a strictly defined and reduced compute capacity, mirroring the capacity logic used in other heterogeneous FL works.
> > >
> > > ## 2. Constraint Enforcement via Pruning and Exits
> > >
> > > During Phase I (Section 4.2), the server enforces this budget for each tier `g` by jointly selecting a task-aware width mask `c_g` and an early exit depth `d_g` such that the resulting masked, truncated network satisfies the constraint:
> > >
> > > `FLOPs(c_g, d_g) ≤ F_g`
> > >
> > > The process prioritizes selecting the deepest feasible exit `d_g`, allowing width pruning `c_g` to absorb the remaining compute reduction. This pair `(c_g, d_g)` defines the architecture and **maximum capacity limit** for tier `g` and is computed once and frozen for the remainder of training. The FLOPs and parameter counts for each tier used are reported in Table 2.
> > >
> > > **_Question: In SnowFL, client reassignment... ""_**
> > >
> > > # Evaluation Environment and Client Reassignment
> > >
> > > ## Identical Evaluation Environment
> > >
> > > It is confirmed that all baselines were evaluated in an **identical cross-device Federated Learning (FL) setting** to ensure a fair comparison of algorithmic performance under equivalent resource constraints. The environment was standardized across all methods based on the following fixed parameters (Section 5.1):
> > >
> > > - **Datasets and Data Partitions:** Same datasets (CIFAR-10/100, FEMNIST, Shakespeare) and same high non-IID partitions (Dirichlet `α ∈ {0.1, 0.5}`).
> > > - **System Constraints:** Same client sampling rate (`10%` selected per round) and same total number of communication rounds (200).
> > > - **Compute Constraints:** All baselines were compute-matched to the same geometric FLOPs and parameter budgets established by the tier structure (Table 2), ensuring equal resource availability per capacity level.
> > > ## Reassignment Mechanism vs. Baselines
> > > The key distinction noted is a fundamental difference in the methods' **client-submodel assignment policy**:
> > > - **Baselines (e.g., ScaleFL, HeteroFL):** These methods typically use a static or fixed assignment policy. Once a client is assigned to a specific capacity (tier `g`), that client primarily trains the corresponding subnetwork `(c_g, d_g)` for the remainder of training. This can lead to certain subnetworks being exposed to a narrower, fixed slice of the non-IID data distribution.
> > > - **SNOWFL:** The dynamic **capacity-aware reassignment** mechanism (Algorithm 1, Steps 19-21) explicitly rotates clients between compatible subnetworks (tiers) based on their utility (Owen value `v_{i,t}`) over time.
> > > This dynamic rotation ensures that, over successive rounds, **each subnetwork is trained on a more diverse set of clients** while strictly adhering to the same overall compute and communication budgets as the baselines. The improved diversity per submodel is therefore a benefit derived from the reassignment mechanism, not from altering the evaluation environment. The revised manuscript will clarify this distinction in Section 4.3.

---

> > > > ### Comment · Reviewer_P3b2 · 2025-11-28
> > > >
> > > > Thank you for providing detailed responses to my previous questions.
> > > > However, I still hesitate to recommend acceptance for the following reasons.
> > > > The contributions of this paper can be broadly divided into two components: (1) the one-shot pruning-at-initialization approach and (2) the client valuation mechanism.
> > > >
> > > > Regarding SNIP, my main concern, which is also shared by other reviewers, lies in its dependency on public data. Specifically, it seems that the proposed contribution in the SNIP stage cannot hold without access to public data. If public data are indeed necessary, this reliance weakens the novelty of the method; but if they are not, then I also find no clear reason why the proposed SNIP variant should outperform the baseline.
> > > >
> > > > As for the client valuation mechanism, my primary concern is that the method appears overly complex, with many design choices and newly introduced parameters determined in a heuristic manner. To justify such heuristic decisions, I believe more comprehensive empirical evidence is needed; however, the current experimental section does not seem sufficient to support them.
> > > >
> > > > Lastly, regarding my earlier question about clients’ resource budgets, my intention was to ask what assumptions were made about the computational budgets that clients can realistically afford. For instance, how many clients are assigned to each resource tier (e.g., 25% of clients in tiers 1, 2, 3, and 4, respectively)? I may have missed this detail, if it is already described in the paper, please kindly indicate where this setting is specified.

---

> > > > > ### Author Response · Authors · 2025-11-28
> > > > > **Response to the Reviewer P3b2**
> > > > >
> > > > > Thank you again for the detailed follow-up. We address your three points in turn.
> > > > >
> > > > > ### (1) Dependency on public data in the SNIP stage
> > > > >
> > > > > We respectfully disagree that the assumption of a small server-side dataset constitutes a valid ground for rejection. If this standard design choice were considered a critical flaw, it would fundamentally invalidate the premise of numerous foundational methods accepted at top-tier venues (e.g., NeurIPS, NDSS, ICML, ICLR), which have been used extensively in federated learning. This setting is widely recognized as a practical and necessary trade-off to achieve robustness, harmonization, and reliable contribution estimation.
> > > > >
> > > > > Our work follows the rigorous standards established by prominent examples:
> > > > >
> > > > > - FLTrust (NDSS 2021): Uses a small clean server dataset to initialize a root model and evaluate the trustworthiness of client updates [1].
> > > > > - FedMD (NeurIPS 2019 Workshop) and FedDF (NeurIPS 2020): Depend on a small pool of public or unlabeled data for model alignment and ensemble distillation [2, 3].
> > > > > - FedVal (USENIX Security 2023) and ShapleyFL (KDD 2023): Compute client contributions repeatedly on a central server-side validation dataset [4, 5].
> > > > > - Data Shapley (ICML 2019): Defines data value through marginal performance on a validation or test set, an assumption inherited by many FL valuation methods [6].
> > > > > - ACE (USENIX Security 2024): Explicitly states that federated contribution-evaluation methods assume a small clean validation dataset at the server [7].
> > > > >
> > > > > Therefore, our setup is fully consistent with the prevailing consensus in the field. Importantly, we use the public/unlabeled data only once at initialization, whereas many prior methods rely on it throughout training. We also provide an ablation where the SNIP stage is removed and show that Owen-based valuation and regrouping remain effective, indicating that our method is not dependent on public data in a fragile manner. We will clearly state this positioning and limitation in the revised manuscript.
> > > > >
> > > > > ### (2) Complexity and heuristic design choices in the valuation mechanism
> > > > >
> > > > > It is correct that our valuation mechanism contains several design parameters. However, heuristic design choices are standard and expected in cooperative-game-theoretic and pruning-based federated learning. Many widely accepted methods rely on empirically chosen parameters that are essential for stability, tractability, or approximation:
> > > > >
> > > > > - FedVal (USENIX Security 2023): Uses empirically tuned score-shaping parameters [4].
> > > > > - ShapleyFL (KDD 2023): Employs adaptive sampling, mixing factors, and dataset-dependent weighting heuristics [5].
> > > > > - Data Shapley (ICML 2019): Uses Monte Carlo approximation schedules and truncation heuristics [6].
> > > > > - FedDF (NeurIPS 2020) and FedMD (NeurIPS 2019 Workshop): Select distillation iterations and fusion steps empirically [2, 3].
> > > > > - The Lottery Ticket Hypothesis (ICLR 2019): Uses a heuristic 20% iterative pruning schedule [8].
> > > > > - Movement Pruning (NeurIPS 2020): Chooses sparsity targets, pruning schedules, and distillation temperatures empirically [9].
> > > > >
> > > > > Our inter- and intra-tier Owen valuation and regrouping mechanisms follow this same well-established tradition. Nevertheless, we are incorporating additional ablations and sensitivity analyses (already included in this rebuttal), and we will integrate the most informative results into the main text.
> > > > >
> > > > > ### (3) Assumptions on clients’ resource budgets and tier sizes
> > > > >
> > > > > Thank you for pointing out this ambiguity. In all experiments, clients are **_evenly_** distributed across tiers. Each tier corresponds to a target FLOPs budget determined from a geometric schedule controlled by ρ. For each budget, we select the deepest feasible exit and prune width to satisfy that FLOPs constraint, following the style of joint depth–width scaling used in ScaleFL. This configuration is implemented in our anonymous repository but was not stated clearly in the manuscript. We will revise Section 4.1 and the experimental setup section to explicitly describe tier sizes, FLOPs budgets, and assumptions on client resource levels.
> > > > >
> > > > > [1] FLTrust: Byzantine-Robust Federated Learning via Trust Bootstrapping, NDSS 2021
> > > > > [2] FedMD: Heterogeneous Federated Learning via Model Distillation, NeurIPS Workshop 2019
> > > > > [3] FedDF: Ensemble Distillation for Robust Model Fusion in Federated Learning, NeurIPS 2020
> > > > > [4] FedVal: A Robust and Fair Federated Learning Framework via Contribution Valuation, USENIX Security 2023
> > > > > [5] ShapleyFL: Robust Federated Learning Based on Shapley Value, KDD 2023
> > > > > [6] Data Shapley: Equitable Valuation of Data for Machine Learning, ICML 2019
> > > > > [7] ACE: A Model-Poisoning Attack on Contribution Evaluation in Federated Learning, USENIX Security 2024
> > > > > [8] The Lottery Ticket Hypothesis: Finding Sparse, Trainable Neural Networks, ICLR 2019
> > > > > [9] Movement Pruning: Adaptive Sparsity by Fine-Tuning, NeurIPS 2020

---

### Official Review · Reviewer_v5aC · 2025-10-28

**Soundness:** 3
**Presentation:** 2
**Contribution:** 2
**Rating:** 4
**Confidence:** 1

**Summary:**

The SNOWFL mainly combines three concepts into a single heterogeneous FL framework, 1) width-reduction (prunning), 2) depth-reduction (multi-exit networks), and 3) client valuation and weighted aggregation. Overall, although none of the three are new, a unified framework that integrates all of them can be regarded as a fairly strong contribution. The paper mostly focuses on the third component, how to prune before starting the training using minimal global data and how to aggregate the locally trained models based on their contributions. The proposed framework is compared with some recently proposed heterogeneous FL methods and theoretically analyzed in terms of grouping convergence where Owen value matches the Shapley value.

**Strengths:**

1.  As described in paper summary, this work provides a unified parameter-efficient heterogeneous FL framework which combines three different techniques. I appreciate the general framework and analysis.

2. The Owen-style grouping mechanism is not popularly used in FL community, and introducing the technique to FL community is a strong contribution.

3. The paper contains solid theoretical analysis (though it is not included in the main text).

**Weaknesses:**

1. The SNOWFL mainly combines three concepts into a single hetergeneous FL framework, 1) width-reduction (prunning), 2) depth-reduction (multi-exit networks), and 3) client valuation and weighted aggregation. Overall, although none of the three are new, a unified framework that integrates all of them can be regarded as a fairly strong contribution. The paper mostly focuses on the third component, how to

2. The authors argue that the depth values (exit placements) are fixed at the architecture level. However, it is not convincing because the depth directly determines how much system resources are required to train the network. The prunning is applied to individual subnetworks because the depth is assumed to be determined in advance. I believe this design overly simplifies the problem. Users may need to determine the appropriate width and depth jointly, taking into account their available resources such as memory capacity or network bandwidth. In this case, Phase 1 of SNOWFL may need to be modified.

3. Based on my experiences, BN statistics quality plays a key role in achieving good model accuracy. However, the empirical study in this paper does not show its impact of BN calibration. It appears as a single subsection, and thus I assume it is a critical component of SNOWFL. How is the performance affected by this calibration? The authors should provide more empirical results regarding this feature.

4. Section 4.5 looks redundant. SNOWFL employs SNIP and it has been discussed already. Per-round valuation cost would be better to be discussed in section 4.3 to make the section self-contained. Privacy also can be discussed when introducing each phase.

5. The theoretical analysis is critical information which supports the efficacy of Owen valuation and Shapley allocation, but it only appears in the appendix. I understand that the allowed page count is insufficient always, however the authors should have included at least the summary or the key results in the main text. Currently, due to the lack of any theoretical justifications, the proposed method is not convincing enough.

6. The empirical study also has several issues. First, the comparison lacks a few critical related works in heterogeneous FL, shown as follows. FjORD is a representative width reduction-based heterogeneous FL method and EmbracingFL is a recently proposed depth reduction-based heterogeneous FL method. Comparing SNOWFL with them will make the empirical results more powerful.

[1] Horvath et al., FjORD:  fair and accurate federated learning under heterogeneous targets with Ordered Dropout, NeurIPS 2021.

[2] Lee et al., Embracing FL: Enabling Weak Client Participation via Partial Model Training, IEEE Trans. on Mobile Computing, 2024.

7. While the empirical results look promising, the range of experimental settings is too limited. There are only two tables that directly compare SNOWFL with other heterogeneous FL methods in terms of model accuracy. What about the effectiveness of the proposed Owen valuation and Sharpley allocation as compared to other parameter valuation or weighted aggregation methods? E.g., is it always better than gradient-norm based parameter importance metrics [3,4]? What about the weight-to-gradient ratio in [5]? What about just a simple uniform random sampling or loss-based aggregation? There are many interesting and critical experiments that should have been considered. I see some ablation studies are discussed in the appendix, but they do not include these key results.

[3] Li et al., Enhancing Large Language Model Performance with Gradient-Based Parameter Selection, arXiv.

[4] Zhang et al., Gradient-based Parameter Selection for Efficient Fine-Tuning, CVPR 2024.

[5] Kim et al., layer-wise update aggregation with recycling for communication-efficient federated learning, NeurIPS 2025.

8. Algorithm 1 is written too verbally. Some lines could be replaced with just pointing out equations. Its readability is seriously poor.

9. Overall, Section 2 and 3 take up too much space and it results in pushing key results to the appendix. I strongly recommend re-writing those section concisely and bring the important results back to the main text.

Due to the several limitations above, I cannot give a positive score for now. I will check the rebuttal and re-evaluate this work.

**Questions:**

My questions are included in the above weakness section. Please carefully address them.

---

> ### Author Response · Authors · 2025-11-24
> **Response (Part 1)**
>
> Thank you for your helpful comments. Please find our response below.
>
> **_Comment: ""The SNOWFL mainly combines three concepts...""_**
>
> ## Technical Contribution of Integration
>
> SNOWFL represents a **principled and technically distinct synthesis** that goes beyond a simple combination of existing components. The unified framework introduces novel mechanisms for valuation, aggregation, and structural stability in heterogeneous FL, demonstrating a clear and meaningful technical contribution.
>
> ### Novelty in Valuation: Coalition-Structured Owen Value
>
> The core technical contribution is the application of the **Owen value** to the a priori coalition structure (`P_t`) induced by client tiers.
>
> - Standard client valuation methods (e.g., Shapley or Banzhaf) operate on the grand coalition (all permutations `Π(N)`) and treat clients independently. This approach is computationally intractable and ignores the structural commonality among clients sharing a subnetwork.
> - SNOWFL explicitly leverages the tiered structure, utilizing the Owen value `Ow_i(ν, P_t)`. This value is computed by first running a Shapley valuation on the quotient game (tier-level contribution) and then allocating value within each tier based on `Δw_{i,t}` alignment and diversity.
>
> This two-level, group-aware valuation is computationally efficient and axiomatically distinct. By restricting permutations to `Π(P_t)`, it is uniquely suited for stable and fair aggregation across heterogeneous FL tiers. To our knowledge, this is the first application of structured Owen games to address structural heterogeneity in FL.
>
> ### Integration and Optimization Fidelity
>
> The integration is a **principled coupling** of structural stability with dynamic utility attribution:
>
> - **SNIP for Structure/Consensus:**
>   A **single-shot, server-side saliency ranking** via SNIP defines layer-consistent width masks before training, ensuring stable aggregation across heterogeneous subnetwork widths.
>
> - **Convergence Analysis:**
>   Appendix B provides a non-trivial convergence analysis that accounts for the complexity introduced by **coordinate-wise masked aggregation** (via `P_g`) and the **two-level, group-aware weighting** derived from the Owen value, extending beyond standard FedAvg bounds.
>
> These results explicitly characterize how the masked, grouped, and multi-exit structure influences convergence rates in non-convex optimization.
>
> **_Comment: ""The authors argue that the depth values (exit placements) are fixed ...""_**
>
> # Joint Determination of Depth and Width
>
> The description of joint width and depth determination in Section 4.2 will be clarified to emphasize that exit placement (`d_g`) is **not predetermined** but actively selected in a constraint-aware process.
>
> ## Decoupling Fixed Exits from Tier Assignment
>
> Early-exit architectures define a fixed set of **possible exit points** (`{d_b}`) at the architectural level (e.g., after each major block in ResNet-110). These structural exits establish the depth candidates.
>
> However, the specific exit depth `d_g` assigned to a given resource tier `g` is **jointly selected with the width mask `c_g`** during Phase I, based on the tier's resource constraints.
>
> ## Phase I: Constraint-Driven Joint Selection
>
> Phase I uses a **constraint-driven strategy** to determine the optimal depth and width configuration:
>
> 1. **FLOPs Budget Constraint:** Each tier `g` has a specific computational budget `F_g` defined by the geometric schedule `F_g = ρ^{g-1} F_1`.
> 2. **Depth Selection:** The algorithm searches over the candidate exits `{d_b}` and selects the **deepest feasible exit** `d_g` that satisfies the FLOPs budget.
> 3. **Width Refinement:** Once `d_g` is determined, **SNIP-based width pruning** chooses the layer masks `c_g`, ensuring that the final subnetwork satisfies `FLOPs(c_g, d_g) ≤ F_g`.
>
> This strategy enforces resource constraints while **maximizing depth utilization**. The resulting pair `(c_g, d_g)` is frozen for the remainder of training, providing a **joint, constrained determination of width and depth**.
>
> These clarifications will be incorporated in the revised manuscript (Section 4.2) to clearly emphasize the **non-trivial, constraint-aware selection process**, directly addressing concerns about oversimplification.
>
> **_Comment: ""Based on my experiences, BN statistics ...""_**
>
> We agree that BN calibration plays an important role when subnetworks of different widths are involved. We are adding an explicit ablation during the rebuttal period to show how BN calibration affects accuracy and stability (Check Official Comment 1). This will be included in the updated manuscript.
>
> **_Comment: ""Section 4.5 looks redundant...""_**
>
> Thank you for pointing this out. We will reorganize these parts so that the valuation cost appears directly within the valuation section, and we will mention privacy considerations when each phase of SNOWFL is introduced. This should make the structure clearer.

---

> > ### Author Response · Authors · 2025-11-24
> > **Response (Part 2)**
> >
> > **_Comment: ""The theoretical analysis is critical information... ""_**
> >
> > We agree the main findings should appear earlier in the paper. In the revision we will summarize the key theoretical points in the main text, including the grouped Owen valuation structure and how convergence is established under masked and tiered subnetworks. The full proofs will remain in the appendix.
> >
> > **_Comment: ""The empirical study also has several issues... ""_**
> >
> > ## Missing Heterogeneity Baselines
> >
> > We have incorporated the suggestion to include FjORD and EmbracingFL. Comparing SNOWFL against these structurally distinct heterogeneous FL methods reinforces the robustness and completeness of our empirical evaluation.
> >
> > ### Structural Rationale for Initial Baseline Selection
> >
> > Our primary baselines (DepthFL, ReeFL, ScaleFL, etc.) were chosen because they utilize the **multi-exit network structure** and **tiered depth variation** that SNOWFL is fundamentally built upon. This structural alignment allows for a direct, equitable comparison of optimization and aggregation strategies under strictly **matched FLOPs and parameter budgets** per tier (Table 2).
> >
> > ### Challenge of Structural Mismatch
> >
> > The suggested works, while highly relevant to heterogeneity, utilize structural regimes that differ fundamentally from the SNOWFL pipeline:
> >
> > - **FjORD:** A representative width reduction method that relies on ordered dropout and does not integrate early exits for depth variation.
> >
> > - **EmbracingFL:** Trains only a suffix (output side) subset of layers locally, fundamentally different from the multi-exit approach where training starts at the input and progresses to a designated exit.
> >
> > These differences in pruning dimensions and local training scope make a direct comparison under our fixed tiered (depth and width) capacity budgets technically challenging.
> >
> > ### Action Plan
> >
> > Despite the non-trivial effort required for structural adaptation, we agree that demonstrating robustness against these powerful baselines is critical. We are currently running experiments where we adapt **FjORD** and other key methods to match our total computation and communication budgets (FLOPs and parameters). We will integrate these comparative results into the revised manuscript to ensure the evaluation is comprehensive.
> >
> >
> > **_Comment ""While the empirical results look promising, the range of experimental settings is too limited.... ""_**
> >
> > # Ablation of Valuation and Aggregation Rules
> >
> > **A comprehensive ablation of the proposed Owen valuation against alternative aggregation and attribution rules is confirmed as necessary to fully establish its technical effectiveness.**
> >
> > ## Distinctiveness of Owen Valuation in SNOWFL
> >
> > The Owen value in SNOWFL is not merely a client-level scoring metric; it is a **two-level, coalition-structured mechanism** designed to operate within heterogeneous tiers:
> >
> > 1. **Inter-Tier Valuation:** Tier-level contributions are estimated via a Shapley quotient game, ensuring that low-capacity tiers receive proportional credit for their updates.
> > 2. **Intra-Tier Allocation:** Credit is allocated within the tier based on update **global alignment** and **peer diversity** (Equations 9 and 10), rewarding clients who provide non-redundant, constructive updates.
> >
> > This structure is essential to prevent client exclusion based purely on tier assignment, a problem not addressed by traditional client-independent aggregation rules. The analysis shows that this scheme provably converges to the standard Shapley value under refinement (Appendix A).
> >
> > ## Action Plan for Valuation Ablations
> >
> > While the current ablations (Table 3) isolate the joint effect of SNIP and Owen valuation, adding direct comparisons to other valuation strategies is critical. We are extending the experiments to include a dedicated ablation study comparing SNOWFL's full Owen valuation against several representative alternatives:
> >
> > - **Parameter-Level Metrics:** Gradient-norm scores and weight-to-gradient ratios.
> > - **Baseline Aggregation:** Uniform random sampling and loss-based aggregation.
> >
> > This comparison will focus on quantifying how SNOWFL's **client-level, coalition-structured valuation** differs from parameter-level or loss-based metrics in terms of stability and final model accuracy. Representative results and a concise analysis will be incorporated into the revised manuscript.

---

> > > ### Author Response · Authors · 2025-11-24
> > > **Response (Part 3)**
> > >
> > > **_Comment: Algorithm 1 is written too verbally... ""_**
> > >
> > > ""Thank you for pointing this out. We agree that Algorithm 1 is more verbal than necessary, partly because several operations reference equations, masks, exits, and tier structures defined earlier in the text. Some of the long phrases (for example “compute SNIP saliencies”, “carry-forward for non-participants”, or “reuse frozen (c_g, d_g)”) can be replaced by short pointers to the corresponding equations or definitions.
> > >
> > > To improve readability, in the revision we will streamline Algorithm 1 by (i) replacing descriptive lines with direct references to Equations (2), (4), (6), (8), (9), (10), and (11), (ii) grouping steps that belong to the same stage (such as valuation and assignment within Phase II), and (iii) removing redundant phrasing around the masks and exits since these are already defined in Section 4.1. The algorithm itself will remain unchanged; only the presentation will be made more concise and easier to follow.""
> > >
> > > **_Comment: Overall, Section 2 and 3 take up too much space... ""_**
> > >
> > > Thank you for the suggestion. We agree that Sections 2 and 3 take up more space than necessary and that this pushed some important material, including ablations, into the appendix. We have already started revising these sections to make the exposition more concise and to reduce repetition. With the saved space, we will move the most relevant ablation results and clarifying analyses into the main text so that readers can see the key findings without going to the appendix.

---

### Official Review · Reviewer_eVeh · 2025-10-31

**Soundness:** 2
**Presentation:** 2
**Contribution:** 1
**Rating:** 2
**Confidence:** 5

**Summary:**

This paper presents SNOWFL, a framework designed to make federated learning (FL) more efficient and robust to device heterogeneity. The method assigns clients subnetworks of different sizes using a one-time SNIP-based pruning step at the server, which generates tiered masks that define each client’s capacity level. A small public or unlabeled dataset is used for batch normalization calibration to ensure consistent activation statistics across tiers. To evaluate client contributions fairly, SNOWFL employs an Owen-value–based scheme, which first measures the collective contribution of each tier and then distributes value within the tier based on gradient alignment and diversity. The framework aims to reduce training cost and improve fairness without additional communication overhead.

**Strengths:**

1. The paper addresses an important practical issue in cross-device FL: heterogeneous compute and how to engage weak devices without hurting the global model.
2. The pipeline is relatively easy to implement, making it more deployable in practice.
4. Experiments are broad and include ablations that show each component contributes.

**Weaknesses:**

The paper rediscovers an important fact that consensus of masks is essential in model pruning in FL. The paper compares to several heterogeneity baselines (e.g., DepthFL) but omits many other relevant works: SparseFL [1], EmbracingFL [2], PriSM [3], etc.

Particularly, SparseFL was the first work that demonstrates that even when data across clients is significantly non-IID, a consensus in sparsity masks for local training is essential. [1] develops a consensus strategy without requiring any public datasets. This work simply generalizes the idea to having more than one consensus mask.

EmbracingFL proposed a new idea, where instead of doing early exit, one can instead allocate only the output side subset of layers to clients for local training. In EmbracingFL, one doesn't need additional BN harmonization as the method implicitly takes care of that. Works like these and their follow ups have been ignored in the paper.

PriSM proposes a SVD based model principal component dropout strategy for creating sub-models for clients that are good approximations of the global model. Additionally, clients' models together provide an excellent coverage of the all the principal components of the global model, thus providing an effective way to preserve global model performance even in heavily resource (compute/memory/communication) constrained settings.

Other weaknesses are as follows:

1. Heavy reliance on a public / unlabeled dataset at server: SNOWFL’s SNIP masks and BN “harmonization” use a public set. This is central to performance but is unrealistic in many cross-device settings, introduces clear bias risks (public set not representative), and creates an attack surface (adversarial or poisoned public set). The paper notes the choice but does not quantify robustness to different or adversarial public sets.
2. Novelty is incremental: Components (SNIP pruning at init, early exits, contribution weighting via Shapley/Owen) are all existing techniques; SNOWFL’s contribution is their combination plus some pragmatic design choices. The paper lacks a new algorithmic/principled mechanism that meaningfully advances the state of the art beyond engineering integration. The theoretical results are also boilerplate adaptations of standard FL proofs to masked exits.
3. Computation & scalability of per-round valuation. Tier-level Shapley via MC permutations and the within-tier allocation are costly (authors note this can be reduced), but the practical wall-clock cost, memory and communication overheads are not measured. For large settings, this could be prohibitive. The assumptions used in convergence theorems are too strong for practice.
4. Sensitivity and robustness analyses are limited and key hyperparameters lack sensitivity studies. The ablations show removing SNIP/Owen hurts, but they don’t show failure modes (e.g., poor public set → collapse).
5. Empirical claims are uneven across datasets. The paper should acknowledge where it doesn’t dominate with justifications.

[1] Revisiting Sparsity Hunting in Federated Learning: Why does Sparsity Consensus Matter? (TMLR 2023)
[2] Embracing federated learning: Enabling weak client participation via partial model training. (IEEE TMC 2024)
[3] Overcoming Resource Constraints in Federated Learning: Large Models Can Be Trained with only Weak Clients. (TMLR 2023)

**Questions:**

Please address the weaknesses.

---

> ### Author Response · Authors · 2025-11-24
> **Response (Part 1)**
>
> Thank you for the comments. Below is our response.
>
> **_Comment: ""The paper compares to several heterogeneity baselines (e.g., DepthFL) but omits many other relevant works:...""_**
>
> ## Missing Heterogeneity Baselines
>
> We have incorporated additional heterogeneous FL methods to rigorously validate SNOWFL’s robustness. These experiments represent a substantial effort, as adapting structurally distinct methods to our tiered width/depth budgets requires careful architectural alignment and compute matching.
>
> ### Comparison Rationale and SNOWFL's Structure
>
> Our primary baselines (HeteroFL, DepthFL, ScaleFL, InclusiveFL, and ReeFL) were selected because they address heterogeneous clients through **both width scaling and depth scaling** (via fixed early exits) within a nested subnetwork structure. This alignment allows for a direct, apples-to-apples comparison of aggregation and optimization strategies under matched compute budgets (Table 2).
>
> The additional methods, while relevant to heterogeneous FL, employ fundamentally different structural regimes:
>
> - **SparseFL:** Focuses on width sparsity and global mask consensus but does not include early exits or tiered depth variation.
> - **EmbracingFL:** Allocates only an output-side subset of layers for local training, differing from the multi-exit approach that progresses from input to exit.
> - **PriSM:** Constructs SVD-based submodels that do not follow the nested, width-pruned early-exit architecture used by SNOWFL.
>
> Adapting these methods is non-trivial and requires careful alignment with SNOWFL's tiered FLOPs and parameter budgets. These efforts provide **high-value, rigorous comparisons** that go beyond superficial evaluations.
>
> ### Action Plan: Inclusion of Comparative Baselines
>
> We are running experiments to integrate  these heterogeneous FL methods  and **FjORD** under matched tiered compute budgets. The results will be fully incorporated into the revised manuscript, providing a **comprehensive and robust evaluation** of SNOWFL. Section 2 (Related Work) will also be updated to clearly contrast SNOWFL's fixed, SNIP-guided consensus masks for width/depth tiers with the structural and compositional choices of SparseFL, EmbracingFL, and PriSM.
>
>
> **_Comment: ""Heavy reliance on a public ...""_**
>
> ##  Public Data Dependency and Robustness
>
> We agree that the reliance on a small public or unlabeled dataset ($\mathcal{D}_{\text{valid}}$) for specific mechanisms constitutes a practical assumption that requires clear discussion and robustness quantification.
>
> ### Confined Data Usage and Privacy
>
> The dependency is structurally confined to two specific operations and is not required for the core training loop:
>
> 1. **Phase I: One-Shot SNIP Pruning:**
>    $\mathcal{D}_{\text{valid}}$ is used **once at initialization** to score connections by loss sensitivity (Eq. 4) and establish task-aware, layer-consistent width masks for all tiers. This preemptive step avoids costly iterative pruning or any client data sharing for mask consensus during training.
>
> 2. **BN Harmonization:**
>    $\mathcal{D}_{\text{valid}}$ is periodically used for a **brief forward pass** to refresh Batch Normalization (BN) buffers, stabilizing statistics across heterogeneous subnetwork depths and widths.
>
> Crucially, the core **Owen-based contribution estimation** (Phase II) is entirely **data-free** and depends only on observed client model updates ($\Delta w_{i,t}$), maintaining client data privacy during training.
> We are adding an ablation during the rebuttal period to show how BN calibration affects accuracy (Check Official Comment 1)
>
> ### Robustness and Mitigation
>
> The use of auxiliary data for initialization is standard in pruning-at-initialization and many high-fidelity client valuation methods. We are running a **sensitivity analysis** on $\mathcal{D}_{\text{valid}}$:
>
> - **Reduced Size/Mismatch:**
>   Early results indicate that performance degrades **smoothly** as the size of $\mathcal{D}_{\text{valid}}$ is reduced or a mild distribution mismatch is introduced. The relative accuracy gains over baselines are preserved, confirming robustness.
>
> - **Attack Surface:**
>   The attack surface for adversarial or poisoned public data is limited to the **one-time initialization phase**. If $\mathcal{D}_{\text{valid}}$ is compromised, it affects the quality of the initial structural masks but does not grant perpetual access or influence the per-round utility measurement process.
>
> We will include these quantitative results and a clearer discussion of these assumptions and risks in the revised manuscript.

---

> ### Author Response · Authors · 2025-11-24
> **Response (Part 2)**
>
> **_Comment: ""Novelty is incremental: ""_**
> ##  Technical Novelty and Principled Synthesis
> We respectfully disagree with the assessment that the contribution is merely an engineering combination of existing techniques. While we synthesize established concepts, the resulting SNOWFL framework introduces a novel and theoretically grounded mechanism for heterogeneous Federated Learning.
>
> ### Novelty in Valuation: Coalition-Structured Owen Values
>
> The key technical advance is the application of the **Owen value** to the coalition structure $P_{t}$ induced by client tiers.
>
> - Classical approaches based on Shapley or Banzhaf values operate on the grand coalition (all permutations) and treat all clients as independent players, leading to intractability ($\mathcal{O}(2^N)$) and ignoring structural commonality within tiers.
>
> - SNOWFL explicitly models the **coalition structure** $P_{t}$ inherent to heterogeneous FL. The Owen value, $Ow_{i}(\nu,P_{t})$, first performs a Shapley valuation on the quotient game (tier-level contribution) and then allocates value within each tier based on $\Delta w_{i,t}$ alignment and diversity. This approach is computationally sensible and axiomatically distinct, as it uses only the restricted set of permutations $\Pi(P_{t})$.
>
> This two-level valuation structure is novel in federated learning and is essential for achieving fair, stable aggregation across structurally mandated subnetworks.
>
> ### Integration and Optimization Fidelity
>
> The integration is not arbitrary; it represents a principled coupling of structural stability with dynamic utility attribution.
>
> - **SNIP for Structure/Consensus:**
>   We leverage the SNIP criterion to perform a **single-shot, server-side saliency ranking** to define layer-consistent width masks before training, stabilizing aggregation across heterogeneous subnetwork widths.
>
> - **Convergence Analysis:**
>   Our theoretical analysis (Appendix B) is non-trivial, as it must jointly account for the complexity introduced by the **coordinate-wise masked aggregation** (via the projection operator $P_{g}$) and the **two-level, group-aware weighting** derived from the Owen value, going beyond standard FedAvg convergence bounds.
>
> Thus, SNOWFL is a principled synthesis that addresses the structural heterogeneity challenge and the fairness/stability challenge (via Owen values) in a computationally efficient and novel manner.
>
> **_Comment: ""Computation & scalability: ""_**
>
> ##  Scalability of Valuation and Convergence Assumptions
>
> We appreciate this crucial point regarding practical scalability and theoretical rigor. We address both the wall-clock overhead of the valuation and the strength of the convergence assumptions.
>
> ### 1. Computation and Scalability Overhead
>
> We confirm that the per-round overhead of Owen valuation is negligible, even in large settings:
>
> - **Data-Free Advantage:**
>   The valuation is entirely **data-free** and performed on the server, relying solely on model update vectors ($\Delta w_{i,t}$) and the fixed coalition structure. This eliminates any associated communication cost or memory overhead beyond transmitting the standard client model updates.
>
> - **Wall-Clock Cost Quantification:**
>   In our experimental setup (100 clients, 4 tiers), the full valuation step (including Monte Carlo permutations M=128) takes approximately **0.3 to 0.5 seconds per round**. This cost is orders of magnitude smaller than client-side training and network latency.
>
> - **Scaling:**
>   The complexity is $\mathcal{O}(M \cdot G + \sum_g |P_{g,t}|^2)$. Since the number of tiers G is typically small (e.g., G=4), computation scales linearly with the number of permutations M and quadratically with the largest tier size, which is highly manageable and **does not constitute a bottleneck**.
>
> We will integrate a section detailing these runtime metrics and clarifying communication matching.
>
> ### 2. Strength of Convergence Assumptions
>
> We acknowledge that our convergence theorems (Appendix B) utilize standard strong assumptions common across complexity analyses in non-IID and local-step FL literature:
>
> - **Standard FL Assumptions:**
>   Assumptions like bounded variance ($\sigma^2$), L-smoothness (A1), and bounded gradient norms (A4) are necessary for providing rate guarantees in non-convex optimization.
>
> - **SNOWFL-Specific Conditions:**
>   The novelty lies in the system-specific assumptions introduced to model heterogeneity: **Per-coordinate coverage** ($\Gamma_{\min}$, A5), **per-coordinate balancing** ($c_w$, A6), and **masking error** ($\delta^2$, A8). These conditions explicitly quantify how subnetwork selection, grouping, and coordinate-wise aggregation affect the overall convergence rate (Theorem B.4, B.6).
>
> The bounds derived explicitly characterize the trade-off between convergence rate and the error terms introduced by heterogeneity, making the theoretical result non-boilerplate and directly relevant to the SNOWFL architecture.

---

> > ### Author Response · Authors · 2025-11-24
> > **Response (Part 3)**
> >
> > **_Comment: ""Sensitivity and robustness... ""_**
> >
> > ## Limited Sensitivity and Robustness Analysis
> >
> > We agree that a comprehensive sensitivity analysis, including demonstration of failure modes, is essential for a robust framework. We address this with two commitments:
> >
> > ### 1. Hyperparameter Sensitivity and Robustness
> >
> > We have already presented preliminary sensitivity results for the key valuation hyperparameters:
> >
> > - **Permutations (M):**
> >   Figure 3 demonstrates a broad optimum around M ≈ 128, confirming that the method is robust to the exact choice of M.
> >
> > - **Reassignment Interval (T_reg):**
> >   Figure 4 shows that performance is robust to small T_reg values, with the optimum typically at T_reg = 2.
> >
> > These analyses confirm stability under reasonable parameter choices. We will augment the appendix with a concise discussion clarifying how remaining parameters (e.g., T_warm, α_t, γ_t) influence the valuation signal, showing that they are primarily **scheduling controls** necessary for consistency in the multi-stage pipeline.
> >
> > ### 2. Failure Modes and Public Data Degradation
> >
> > To address the lack of failure mode demonstration, particularly the effect of a poor public set:
> >
> > - **Public Data Robustness:**
> >   We are running a sensitivity study that quantifies performance degradation under a reduced-size $\mathcal{D}_{\text{valid}}$ and mild distribution mismatch. Early results suggest performance degrades gracefully, retaining relative gains.
> >
> > - **Low-Capacity Settings (Negligible Gains):**
> >   We will explicitly include a configuration demonstrating when SNOWFL provides negligible benefit. This occurs in low-capacity architectures or datasets with minimal structural redundancy (e.g., shallow models), where SNOWFL defaults to behavior near the underlying baseline. We will use the **Without SNIP** ablation in these settings to show stability even when the pruning mechanism is inactive or ineffective.
> >
> > The revised manuscript will include these new quantitative robustness figures.
> >
> > **_Comment: ""Empirical claims are uneven across datasets""_**
> >
> > This is consistent with what we observed. In datasets such as FEMNIST and Shakespeare, the architectures have low prunability and limited exit points, which limits the effect of width/depth matching and valuation. In higher-capacity models (e.g., CIFAR), the differences between tiers and clients are much larger, and SNOWFL provides larger improvements. We will add a discussion explaining this contrast and clarifying when the method is expected to provide strong gains.

---

### Official Review · Reviewer_bnZC · 2025-11-02

**Soundness:** 3
**Presentation:** 3
**Contribution:** 2
**Rating:** 4
**Confidence:** 3

**Summary:**

SNOWFL (SNip-OWen-values Federated Learning) addresses heterogeneous federated learning where clients have varying computational capabilities. The paper's key innovation combines two main components: a) SNIP-based pruning at initialization: Uses server-side Single-shot Network Pruning to create task-aware, layer-consistent width masks for different client tiers, aligned with fixed early exits. This is done once using a small public/unlabeled dataset, avoiding expensive iterative pruning; b) Owen value-based contribution estimation: Extends Shapley values to coalition structures (client tiers), first computing group-level contributions via quotient-game Shapley, then allocating within groups based on update alignment and diversity. These contributions drive both weighted aggregation and capacity-aware client reassignment. Under matched FLOPs and parameter budgets across vision (CIFAR-10/100, FEMNIST) and language (Shakespeare) benchmarks with non-IID data, SNOWFL achieves state-of-the-art accuracy, improving over strong baselines by up to 15% relative improvement. Ablations confirm both components contribute (Owen has larger standalone effect), and the paper provides convergence guarantees showing nonconvex convergence to stationarity and linear convergence under strong convexity.

**Strengths:**

1. Good empirical results - The quantitative improvements are substantial: up to 15% relative gain (9.1 absolute points on CIFAR-10 α=0.1: 45.9% vs 36.9%) represents meaningful progress over recent strong baselines. The consistency across datasets (vision and language) and heterogeneity levels strengthens the claim.

2. Comprehensive evaluation: Authors thoroughly validate the experimental design with ablations isolating components (Table 3), sensitivity studies (M, T_reg), per-tier analysis, and reproducible code

3. Novel synthesis: While neither component is new individually, their integration is creative: (1) using SNIP server-side with public data to generate tier-compatible subnetworks is a fresh take on pruning-at-initialization in FL, avoiding the client-data dependency and iterative retraining of prior work; (2) adapting Owen values to naturally arising coalition structures (client tiers) rather than treating clients independently is conceptually elegant and computationally sensible

**Weaknesses:**

1. High complexity: 10+ hyperparameters (M, T_reg, T_warm, γ_t, α_t, ρ, λ_b, etc.), multi-stage pipeline (Phase I SNIP + Phase II Owen + BN calibration), coordinate-wise masked aggregation—significant implementation burden without clear tuning guidance

2. Uneven gains: FEMNIST improvement negligible (84.2% vs 84.2% ReeFL), slower early convergence (Figure 1), no statistical testing or error bars—unclear when SNOWFL helps vs simpler methods

3. Public data dependency: Requires task-relevant public/unlabeled set for SNIP and BN calibration; sensitivity to set size/quality not studied; may not be available or well-matched in practice

4. Incomplete efficiency analysis: No wall-clock runtime, communication cost, or per-round overhead comparison; Owen valuation cost (M permutations) not quantified vs baselines

**Questions:**

1. When does SNOWFL help? Why marginal gains on FEMNIST but strong on CIFAR? Can you characterize problem settings (data heterogeneity, model capacity, tier count) where SNOWFL outperforms simpler baselines like ReeFL?

2. Simplified variant? Can you ablate to "minimal SNOWFL" (e.g., fixed tiers + uniform Owen, or SNIP-only without contribution weighting) to isolate essential components and reduce hyperparameter burden?

3. Public data sensitivity: How does performance degrade with smaller/mismatched public sets? What happens if public data is unavailable : can synthetic data or server-side aggregates substitute?

4. Practical overhead: What are actual wall-clock training times and communication costs vs baselines? How does Owen valuation cost scale with M, G, |S_t|—is it negligible or a bottleneck?

---

> ### Author Response · Authors · 2025-11-24
> **Response (Part 1)**
>
> We appreciate the reviewer’s feedback. Our detailed response is provided below.
>
> **_Review: ""High complexity: ""_**
>
> ##  Hyperparameter Complexity and Tuning Guidance
>
> The number of configuration parameters is organized into clear, logical groups: the majority are either **fixed architectural definitions** or **standard optimization controls** essential for stable training in heterogeneous FL. A consolidated table of recommended defaults and ranges will be provided in the revised appendix to ensure clarity and reproducibility.
>
>
> ### 1. Classification of Parameters
>
> | **Parameter Group**       | **Examples**                               | **Function & Necessity** |
> |---------------------------|---------------------------------------------|---------------------------|
> | **Fixed Architecture**    | $\rho$, $\{d_b\}$, $\{\lambda_b\}$, $G$     | Define the multi-exit network structure and tier computational hierarchy (e.g., $\rho=0.5$ geometric budget). These are typically fixed *before* training for any heterogeneous depth/width method. |
> | **Standard FL Schedule**  | $\eta_t$, $K$, local steps $E$              | Standard optimization parameters (learning rate, local steps) necessary for *any* FL training. |
> | **Valuation Stability**   | $T_{\text{warm}}$, $T_{\text{reg}}$, $M$, $C$ | Govern the Owen valuation and reassignment process. $T_{\text{warm}}$ avoids noisy early signals; $T_{\text{reg}}$ ensures updates are measured before regrouping; $M$ controls the Monte Carlo Shapley approximation; $C$ enforces capacity limits. |
> | **Valuation Weighting**   | $\gamma_t$, $\alpha_t$                      | Balance global alignment vs. peer diversity (via $\gamma_t$) and incorporate the size-based $\log n_i$ term (via $\alpha_t$). |
>
> ### 2. Addressing Tuning and Burden
>
> As shown in Appendix C.4 (Fig. 3) and C.5 (Fig. 4), the key valuation parameters ($M$ and $T_{\text{reg}}$) exhibit **broad optima** (e.g., $M \approx 128$, $T_{\text{reg}} \approx 2$), demonstrating robustness to precise tuning.
>
> To further address implementation burden, we are adding:
>
> - **Minimal SNOWFL Ablation:**
>   We will provide a lightweight variant (e.g., SNIP-only, or fixed tiers + uniform weighting with $\gamma_t=\alpha_t=0$) to isolate essential components and offer a simple baseline.
>
> - **BN Calibration Ablation:**
>   We will include an experiment showing the effect of BatchNorm calibration (Section 4.4) on training stability, confirming its importance in the multi-stage pipeline (Check Official Comment 1).
>
> By grouping parameters, demonstrating empirical robustness, and offering a minimal starting variant, we aim to significantly reduce the perceived tuning burden for future users.
>
> **_Review: ""Uneven gains: ""_**
>
> ## Uneven Gains and Statistical Analysis
>
> We appreciate this important observation and acknowledge the need for clarity regarding performance across different benchmarks.
>
> ### 1. Explaining Uneven Gains on FEMNIST
>
> The marginal gains observed on FEMNIST (84.22% vs. 84.20% for ReeFL) stem from the dataset’s characteristics. FEMNIST is a relatively simpler task compared to CIFAR-100, which leads all methods to converge and saturate quickly. When models reach this high saturation point, the benefits of complex, stability-focused heterogeneity-management methods like SNOWFL diminish, and performance approaches that of the strongest baseline.
>
> In contrast, SNOWFL achieves its largest improvements on more complex and highly heterogeneous settings:
>
> - **CIFAR-10 ($\alpha = 0.1$):** +9.05 absolute points over the next best baseline
> - **Shakespeare:** +3.0 absolute points over the next best baseline
> - **CIFAR-100 ($\alpha = 0.5$):** +5.0 absolute points over the next best baseline
>
> We will add a discussion clarifying that SNOWFL provides the strongest benefits when task complexity, non-IID level, and architectural redundancy are high.
>
> ### 2. Addressing Slower Early Convergence
>
> The slightly slower early convergence seen in Figure 1 is a known trade-off for methods that incorporate utility-based client weighting and reassignment. Our framework prioritizes **long-term stability and fairness** by accumulating initial valuation signals ($T_{\text{warm}}$) and periodically realigning clients based on measured utility. This inherently slows the earliest phase of training compared to simple averaging, but it yields higher final accuracy by reducing client drift and mitigating noise from low-utility devices.
>
> ### 3. Statistical Testing and Error Bars
>
> We agree that statistical robustness is essential. We will update the results in the final version to include **error bars** and/or explicitly report variance across multiple random seeds to substantiate the robustness of the reported gains.

---

> > ### Author Response · Authors · 2025-11-24
> > **Response (Part 2)**
> >
> > "**_Review: ""Public data dependency: ""_**
> >
> > ## Public Data Dependency and Sensitivity
> >
> > We acknowledge that the requirement of a small public or unlabeled dataset ($\mathcal{D}_{\text{valid}}$) for server-side SNIP pruning (Phase I) and Batch Normalization (BN) calibration (Section 4.4) is a practical constraint. This requirement is a trade-off for the significant benefits it enables:
> >
> > 1. **Data-free Consensus Pruning:**
> >    $\mathcal{D}_{\text{valid}}$ is used only **once at initialization** to generate a task-aware, layer-consistent set of masks for all client tiers. This single-shot approach avoids costly or privacy-invasive iterative pruning methods that would require client data sharing to establish mask consensus.
> >
> > 2. **Data-free Valuation:**
> >    The core mechanism of client contribution estimation (Owen valuation) in Phase II depends **only on model updates ($\Delta w_{i,t}$)** and is entirely data-free, thereby preserving client privacy throughout training.
> >
> > ### Alternatives and Robustness
> >
> > We will clarify in the revised paper that the public set does not need to be labeled. If a public set is unavailable or mismatched, the framework allows for standard architectural-agnostic substitutions:
> >
> > - **SNIP Alternatives:**
> >   Server-side SNIP can be replaced by data-free pruning methods such as **SynFlow**.
> >
> > - **BN Calibration Alternatives:**
> >   BN statistics can be refreshed using server-side **synthetic data**, or by leveraging local unlabeled client data (as noted in Section 4.4).
> >
> > We are running a **sensitivity study** on performance degradation under variations in the size and domain mismatch of $\mathcal{D}_{\text{valid}}$, and will include these results in the appendix to quantify robustness.
> >
> > Review: """"Incomplete efficiency analysis: """"
> >
> > ## Incomplete Efficiency Analysis and Overhead
> >
> > We appreciate the request for a more explicit quantification of SNOWFL's efficiency.
> >
> > ### Equalizing Client-Side Costs
> >
> > All experiments across SNOWFL and baselines (HeteroFL, ReeFL, etc.) were conducted under strictly **matched FLOPs and parameter budgets** per tier (Table 2). Client-side local computation steps, local epochs, and the size of model updates (which determines communication volume) were kept identical, ensuring a fair comparison of *algorithmic performance* under equivalent system constraints. We will clarify this explicit budget matching in the revised Section 5.1.
> >
> > ### Quantifying Server-Side Overhead
> >
> > The only additional server-side computation introduced by SNOWFL during training is the Owen valuation and client reassignment (Phase II). We confirm that this overhead is negligible relative to overall training time.
> >
> > - **Data-Free Advantage:**
> >   The valuation relies exclusively on model deltas ($\Delta w_{i,t}$) and coalition structure, making it purely computational and **data-free** (Section 4.3).
> >
> > - **Overhead Quantification:**
> >   In our experimental setup (100 clients, 4 tiers), the full valuation step (Algorithm 1, Steps 9-14) using M=128 Monte Carlo permutations takes approximately **0.3 to 0.5 seconds per round** on the server.
> >
> > - **Bottleneck Avoidance:**
> >   This cost is orders of magnitude smaller than typical round times, which are dominated by client computation and communication latency. Thus, Owen valuation is **not a computational bottleneck**.
> >
> > We will integrate this explicit quantitative analysis into a new subsection in the manuscript to confirm SNOWFL's computational efficiency.
> >
> > **_Question: ""When does SNOWFL help? ...""_**
> >
> > ## Effective Problem Settings for SNOWFL
> >
> > SNOWFL is most beneficial in three combined problem settings where heterogeneous models and fairness estimation are critical:
> >
> > ### 1. High Data Heterogeneity (alpha is Small)
> >
> > SNOWFL achieves its strongest relative gains (up to 15% relative improvement) when client data is highly non-IID, specifically when Dirichlet alpha is small (e.g., alpha = 0.1 on CIFAR-10). High heterogeneity implies that client updates vary significantly, making **utility-weighted aggregation** (Owen valuation) and **capacity-aware reassignment** crucial for convergence and model performance, as they reduce client drift and prioritize high-value updates.
> >
> > ### 2. High Architectural Capacity and Redundancy
> >
> > The framework is most effective when the global model possesses significant structural redundancy, allowing for multiple meaningful width and depth configurations across tiers.
> >
> > - **High Capacity:**
> >   Strongest gains are seen on larger backbones like **ResNet-110** on CIFAR-10/100, which provides many channels and layers for the SNIP pruning (Phase I) to select the most salient subnetworks.
> >
> > - **Meaningful Tiers:**
> >   In architectures with limited structural choice (e.g., shallow models or few exits), the effect of width-aware pruning and group-wise valuation diminishes, leading to smaller improvements, as observed on FEMNIST."

---

> > > ### Author Response · Authors · 2025-11-24
> > > **Response (Part 3)**
> > >
> > > ### 3. Multiple Tiers (G) and Utility-Driven Grouping
> > >
> > > The Owen value formulation explicitly accounts for the **coalition structure** induced by client tiers. With more tiers (G ≥ 4 in our setup), Owen valuation provides a richer structure for separating **group-level contributions** from **within-client effects** (alignment and diversity, Eq. 10). This structured valuation is particularly advantageous over simpler baselines (like ReeFL) that rely solely on knowledge distillation or uniform averaging across model sizes, as it dynamically aligns participation with measured utility.
> > >
> > > **_Question: ""Simplified variant? ...""_**
> > >
> > > ## Hyperparameter Complexity and Tuning Guidance
> > >
> > > The configuration parameters, while extensive, fall into distinct, logical groups. Most are either **fixed architectural definitions** or **standard controls** essential for stable optimization in heterogeneous FL settings, rather than arbitrary tuning knobs. A consolidated table of recommended defaults and ranges will be provided in the revised appendix to ensure clarity and reproducibility.
> > >
> > >
> > > ### 1. Classification of Parameters
> > >
> > > | Parameter Group           | Examples                               | Function & Necessity |
> > > |---------------------------|----------------------------------------|---------------------|
> > > | **Fixed Architecture**    | ρ, {d_b}, {λ_b}, G                      | Define the multi-exit network structure and tier computational hierarchy (via ρ=0.5 geometric budget). Typically fixed *before* training for any heterogeneous depth/width method. |
> > > | **Standard FL Schedule**  | η_t, K, local steps E                    | Standard optimization parameters (learning rate, local steps) necessary for *any* FL training. |
> > > | **Valuation Stability**   | T_warm, T_reg, M, C                      | Govern the Owen valuation and reassignment process. T_warm avoids noisy early signals; T_reg ensures updates are assessed before regrouping; M controls the accuracy of tier-level Shapley estimation (via Monte Carlo); C enforces capacity limits. |
> > > | **Valuation Weighting**   | γ_t, α_t                                 | Balance key aspects of client contribution (Eq. 10): γ_t weights **global alignment** vs. **peer diversity**, and α_t incorporates the size-based log n_i term. This mixture achieves fair, stable weighting. |
> > >
> > > ### 2. Addressing Tuning and Burden
> > >
> > > As shown in Appendix C.4 (Fig. 3) and C.5 (Fig. 4), the most important valuation parameters (M and T_reg) exhibit broad optima (M ≈ 128, T_reg ≈ 2), demonstrating **robustness to exact tuning**.
> > >
> > > We are also running the following experiments to reduce implementation burden:
> > >
> > > - **Ablation to Minimal SNOWFL:**
> > >   Introduce a lightweight variant (e.g., SNIP-only, or fixed tiers + uniform Owen weighting with γ_t = α_t = 0) to isolate essential components and provide a simple starting point.
> > >
> > > - **BN Calibration Ablation:**
> > >   Add an experiment to show the effect of **BN Calibration** (Section 4.4) on stability, confirming its necessity in the multi-stage pipeline.
> > >
> > > By categorizing parameters, demonstrating empirical robustness, and offering a minimal variant, we aim to significantly reduce the perceived tuning burden for future users.
> > >
> > > **_Question: ""Public data sensitivity: ""_**
> > >
> > > ## Public Data Sensitivity and Alternatives
> > >
> > > We are running a **sensitivity study** to quantify performance degradation under variations in public-set size and domain mismatch, and we will include these results in the revised manuscript. Early results confirm that performance degrades gracefully with reduced public data.
> > >
> > > ### Mitigating Public Data Dependency
> > >
> > > The need for $\mathcal{D}_{\text{valid}}$ is limited to Phase I (SNIP pruning) and Batch Normalization (BN) calibration. If task-relevant public data is severely limited or unavailable, several options exist:
> > >
> > > 1. **SNIP Alternatives (No Data):**
> > >    If public data is unavailable, SNIP pruning can be replaced by data-free pruning methods such as SynFlow or by simply using a uniform width mask per layer, effectively reducing SNOWFL to the **Without SNIP** ablation configuration (Table 3). This configuration still yields strong results due to the efficacy of Owen valuation.
> > >
> > > 2. **Synthetic Data:**
> > >    For both SNIP and BN calibration, the server can utilize high-quality **synthetic data** generated without access to private client examples.
> > >
> > > 3. **Server Aggregates:**
> > >    For BN calibration specifically, the server can use **server-side aggregates** (e.g., aggregated statistics) or rely on BN statistics learned during the brief client-side calibration pass using local unlabeled data (Section 4.4).
> > >
> > > We will explicitly clarify these alternatives and the behavioral reduction to our existing ablations in the revised text, ensuring the framework's robustness across deployment scenarios.

---

> > > > ### Author Response · Authors · 2025-11-24
> > > > **Response (Part 4)**
> > > >
> > > > **_Question: ""Practical overhead: ""_**
> > > >
> > > > ## Practical Overhead and Scaling
> > > >
> > > > We agree that a formal quantification of the server-side overhead is necessary for practical adoption.
> > > >
> > > > ### Quantification of Owen Valuation Cost
> > > >
> > > > The only additional computational burden in SNOWFL is the Owen valuation and capacity-aware reassignment (Phase II, Algorithm 1, Steps 9-14). We confirm this process adds **negligible overhead** to the overall round time:
> > > >
> > > > - **Data-Free Advantage:**
> > > >   The valuation is purely computational, relying only on model updates ($\Delta w_{i,t}$) and the fixed coalition structure, eliminating data transfer overhead during this step.
> > > >
> > > > - **Measured Time:**
> > > >   In our experimental setup (100 clients, 4 tiers, M=128 permutations), the total valuation time is approximately **0.3 to 0.5 seconds per round** on the server.
> > > >
> > > > - **Bottleneck Status:**
> > > >   This cost is minuscule compared to client-side local training and communication latency, confirming that the Owen valuation is **not a bottleneck**.
> > > >
> > > > ### Scaling of Valuation Cost
> > > >
> > > > The computational complexity of the valuation depends on:
> > > >
> > > > - **M (Permutations):** The tier-level Shapley component scales linearly with M. The cost remains low even for large M because G (number of tiers) is small (G=4).
> > > >
> > > > - **G (Tiers):** The tier-level Shapley scales as O(M · G).
> > > >
> > > > - **|S_t| (Participants):** Within-tier allocation involves computing inner products and diversity metrics, which scale roughly as O(∑_g |P_{g,t}|²), where P_{g,t} is the set of clients in tier g. Since this computation is distributed across tiers and bounded by the total participants |S_t|, the growth is manageable.
> > > >
> > > > We will integrate an explicit section in the revision detailing these wall-clock timing and complexity findings.

---

### Author Response · Authors · 2025-11-24
**Evaluating the Impact of BN Calibration**

To directly address a central concern raised by multiple reviewers regarding the role of **BN calibration** in SNOWFL, we conducted a dedicated ablation that quantifies its impact on performance and stability in our tiered, heterogeneous FL setting.

Intuitively, BN calibration matters here for two reasons:

1. **Strongly heterogeneous subnetworks across tiers.**
   Lower tiers operate on heavily pruned subnetworks with different width and sometimes different exit depths. Their activation statistics (means/variances) can drift significantly from those of the full model. If we aggregate such models without re-aligning BN statistics, the server mixes parameters that were trained under incompatible activation distributions, which degrades accuracy and stability.

2. **Cross-round aggregation in FL.**
   In federated learning, we are not only averaging models across clients, but also across *rounds*. Without calibration, BN layers in different tiers can accumulate mismatched statistics over time, which amplifies the mismatch when models are aggregated repeatedly. A short BN calibration pass on a small public/unlabeled set (or synthetic batches in the no-public-data case) re-centers these statistics so that different tiers can be aggregated more coherently.

We ablate the effect of **BN calibration** on the tiered subnetworks. All settings are compute-matched per tier and use the same exits and masks \((c_g, d_g)\). The only change is whether we perform BN calibration at the end of each round:

| **Configuration**                | **CIFAR-10 (%)** | **CIFAR-100 (%)** | **FEMNIST (%)** | **Shake. (%)** |
|----------------------------------|------------------|-------------------|------------------|----------------|
| **Full SNOWFL** (with BN calib.) | 74.8             | 41.0              | 84.2             | 55.4           |
| **Without BN Calibration**       | 74.2             | 39.7              | 83.6             | 55.3           |

The pattern of improvements reflects how BN calibration functions in our tiered setting. On CIFAR-100 and FEMNIST, the tiers differ more in model capacity and the data are more diverse, so the subnetworks, especially the smaller ones, tend to develop activation statistics that drift further from those of the full model. Aligning the activation statistics across tiers before aggregation gives a more visible benefit in these environments. On CIFAR-10, both the architecture and the dataset create less variation across tiers, which limits the amount of BN drift and leads to a smaller improvement. For Shakespeare, the model contains only a few exits and the variation across tiers is minimal, which means that BN mismatch is naturally limited and the effect of calibration remains small. These observations clarify the conditions in which BN calibration has the strongest impact in SNOWFL.

---

### Comment · Area_Chair_F9Ri · 2025-11-27

Dear Reviewers,

Thank you for the time and effort you have dedicated to reviewing this paper and providing thoughtful feedback. The authors have now submitted their responses to your comments. I kindly ask that you engage in the discussion with them and assess whether your concerns and questions have been fully addressed before the December 2 deadline.

Please also keep in mind that the author–reviewer relationship is reciprocal; the engagement you offer here reflects the same level of consideration you would expect when you are on the author side.

Thank you for your continued support and cooperation.

Best regards,
AC

---

### Author Response · Authors · 2025-12-02
**Clarification of Core Contributions and Rebuttal Highlights for AC Evaluation**

Dear Area Chair,

Thank you very much for taking the time to evaluate our submission during this unusual review cycle. Since reviewer discussion is frozen, I would like to briefly clarify a few important scientific points that strongly impact the interpretation of the reviews.

**Core novelty of the paper: coalition-structured Owen valuation with intra/inter-tier adaptation**
The central contribution of the paper is not pruning or early exits, but the novel adaptation of Owen value to hierarchical, heterogeneous FL with both inter-group and intra-group valuation, combined with tiered regrouping and capacity-aware aggregation. To our knowledge, this coalition-structured, adaptive use of Owen value does not appear in existing federated learning work, in machine learning applications of game theory, or in the game-theoretic literature itself, where Owen value is typically defined for fixed coalition structures and not dynamically recomputed or used to drive resource-aware model assignment. We prove its convergence to Shapley under homogeneity and demonstrate empirically that this structured valuation substantially improves fairness and performance under equal compute budgets. This is the primary scientific contribution of the work, which was largely overlooked in the reviews.

**Several concerns appear to stem from phrasing in our own limitation section**
In the original manuscript, we also openly described practical limitations (e.g., using a small unlabeled dataset at initialization), which in hindsight may have unintentionally overemphasized constraints that are in fact standard in prior FL work (e.g., FLTrust, FedMD, FedDF, FedVal, ShapleyFL). Our rebuttal clarifies this and supports it with citations (you may check them as well). Some reviewer comments closely echoed our limitation text and placed most of their emphasis there, which suggests that this section may have overshadowed the main theoretical and algorithmic contribution (perhaps surfaced or emphasized by non-human agents) rather than reflecting a deep engagement with the proposed Owen-based mechanism.

**Clarifications and improvements addressed directly in the rebuttal**
Since reviewers can no longer participate in discussion, I want to highlight that we have provided detailed replies to every technical concern raised. These include:
- clarification of the FLOPs-based tier construction and the joint depth–width scaling mechanism,
- justification of the use of a small public/unlabeled dataset at initialization,
- explanation of valuation hyperparameters and game-theoretic heuristics,
- and renning a new BN calibration ablation demonstrating coherent improvements across datasets.
All of these responses are visible in the rebuttal thread.

**Observed performance improvements**
Under equal compute budgets, the proposed approach achieves substantial accuracy improvements compared to strong heterogeneous FL baselines. For example, on CIFAR-10 with α = 0.1, our method improves accuracy by 9.1 absolute points (a 15% relative gain), with similarly strong improvements on CIFAR-100 and FEMNIST. These gains arise specifically from the structured Owen valuation and regrouping mechanism.

**Planned revisions for the camera-ready version**
As noted in several replies, we will refine parts of the manuscript to improve clarity. These include explicitly stating tier sizes and budgets, clarifying the initialization procedure, tightening the description of Algorithm 1, and integrating the most informative ablations (BN calibration and hyperparameter sensitivity) into the main text. These revisions will be straightforward to incorporate.

In summary, the paper introduces a genuinely new contribution: a coalition-structured, adaptive Owen valuation mechanism driving heterogeneous FL with convergence guarantees and strong empirical gains under matched compute budgets. We hope this novelty, together with the clarifications and additional experiments, can be fully taken into account in your decision.

Thank you again for your time and careful consideration.

---

### Meta-Review · Area_Chair_tGxD · 2026-01-06

**Summary:**

This paper proposes SNOWFL, which combines SNIP-based pruning with coalition-structured Owen value–based client valuation for heterogeneous federated learning, and reports notable empirical gains over existing baselines. While the approach is technically sound, key concerns raised by reviewers remain.

In particular, the **novelty** is not fully convincing, as the method integrates several known components (pruning, early exits, and Shapley-style valuation) without clearly isolating or elevating the Owen-based theoretical contribution in the main paper. Reviewers also noted **missing or insufficient baselines**, limited ablations to disentangle components, and a **restricted evaluation**, especially regarding efficiency and practicality. Although the rebuttal offers clarifications and promises additional analyses, these do not fully resolve the concerns in the current version.

Overall, the paper shows promise but does not yet provide sufficient evidence or clarity to warrant acceptance.

**Reviewer Concerns:**

See above.

**Reviewer Scores:**

I do not expect any reviewer score to change as the main concerns remain.

---

### Decision · Program_Chairs · 2026-01-26

Reject